# An epitranscriptomic program maintains skeletal stem cell quiescence via a METTL3-FEM1B-GLI1 axis

Jing Wang [1,2,8], Weidong Liu[1,8], Tiantian Zhang[3], Manman Cui[3], Kexin Gao[3], Pengbo Lu[3], Shuxin Yao[3], Ziyan Cao[3], Yanbing Zheng[3], Wen Tian[3], Yan Li[3], Rong Yin[3,4], Jin Hu[3], Guoqiang Han [3,4], Jianfei Liang [5], Fuling Zhou[4], Jihua Chai [1✉] & Haojian Zhang [1,3,4,6,7✉]

## Abstract

Skeletal stem cells (SSCs) maintain the skeletal system via pluripotency and differentiation capacity. However, it remains largely unknown how these cells precisely regulate their function to maintain skeletal organization. Here, we delineate the RNA m⁶A modification landscape across skeletal cell populations in the mouse epiphysis. Our findings show that m⁶A modifications are prevalent in skeletal stem cell and progenitor populations and play critical roles in cell fate determination. Genetic deletion of Mettl3, the core catalytic subunit of the m⁶A-methyltransferase complex, in murine skeletal stem and progenitors impaired bone development, leading to shortened limbs, disrupted growth plate zonation, and decreased bone mass. Moreover, Mettl3 deficiency induced quiescence exit in SSCs, together with compromised self-renewal capacity and differentiation potential. Mechanistically, Mettl3-mediated m⁶A modification reduced mRNA stability of the Cul2-RING E3 ligase complex subunit Fem1b, which subsequently stabilizes Gli1 protein, a key transcription factor of Hedgehog pathway for maintaining SSC identity and function. Thus, we present a comprehensive RNA m⁶A modification landscape of skeletal cell hierarchy and uncover the essential function of epitranscriptomically-regulated proteostasis in maintaining SSCs quiescence and potency.

**Keywords** Skeletal Stem Cell; RNA m⁶A; Quiescence; Proteostasis; FEM1B
**Subject Categories** Chromatin, Transcription & Genomics; Development; Post-translational Modifications & Proteolysis

## Introduction

The skeletal system is composed by multiple tissue types including bone, cartilage, hematopoietic cells, nerves and fibroblasts, which is maintained by distinct stem cells (Greenbaum et al, 2013; Kronenberg,

2003; Li et al, 2022a; Méndez-Ferrer et al, 2010; Mizuhashi et al, 2018; Ono et al, 2014). For instance, hematopoietic stem cells (HSCs) maintain the hematopoietic homeostasis via the capability of self-renewal and multipotency, which are tightly governed by the intrinsic and extrinsic regulatory networks (Cheng et al, 2023; Meng et al, 2023; Yin et al, 2021). Recently, a series of seminal studies using lineage tracing technique, single-cell RNA sequencing (scRNA-seq) and fluorescence-activated cell sorting (FACS) have identified and characterized skeletal stem cells (SSCs) and its downstream progenitors, including pre-bone cartilage and stromal progenitors (pBCSP) and BCSP (Chan et al, 2015; Chan et al, 2018). From a developmental perspective, these studies uncover the skeletal hierarchy, and demonstrate that these populations are essential for bone development, growth and regeneration (Ambrosi et al, 2021; Murphy et al, 2020; Newton et al, 2019; Pandey et al, 2023; Yuan et al, 2024; Yue et al, 2016). However, it remains largely unknown how the skeletal system is precisely regulated.

Epigenetic and epitranscriptomic modulations play key roles in regulating organic development and homeostasis, and stem cell maintenance and function (Jin et al, 2024; Wang et al, 2020). Recently, we established an epigenetic landscape along the developmental trajectory of SSCs in skeletogenesis and identified the key epigenetic factor PTIP in governing the SSCs lineage output (Liang et al, 2024). N6-methyladenosine (m⁶A) is viewed as the most enriched modification in mammalian messenger RNAs (mRNAs) (Dominissini et al, 2012; Meyer et al, 2012). This reversible modification is catalyzed by a multicomponent methyltransferase complex that is composed with two core methyltransferase-like proteins (METTL3 and METTL14) and several regulatory proteins. The demethylases ALKBH5 and FTO are responsible for erasing m⁶A (Jia et al, 2011; Liu et al, 2013; Zheng et al, 2013). It is known that m⁶A plays dynamic roles in various physiological and pathological conditions (Han et al, 2019; Mauer et al, 2016; Winkler et al, 2018; Xiang et al, 2017; Zhou et al, 2015). Similarly, recent studies show that several m⁶A regulators are implicated in bone development and homeostasis (He et al, 2022; Li et al, 2022b; Liu et al, 2022; Wu et al, 2018). However, the m⁶A landscape in skeletal hierarchy especially SSCs and progenitors

[1]State Key Laboratory of Oral & Maxillofacial Reconstruction and Regeneration, Key Laboratory of Oral Biomedicine Ministry of Education, Hubei Key Laboratory of Stomatology, School & Hospital of Stomatology, Wuhan University, Wuhan, China. [2]Department of Stomatology, Renmin Hospital of Wuhan University, Wuhan, China. [3]Frontier Science Center for Immunology and Metabolism, Medical Research Institute, Wuhan University, Wuhan, China. [4]Department of Hematology, Zhongnan Hospital, Wuhan University, Wuhan, China. [5]Key Laboratory of Shaanxi Province for Craniofacial Precision Medicine Research, College of Stomatology, Xi'an Jiaotong University, 710004 Xi'an, China. [6]Taikang Center for Life and Medical Sciences, Wuhan University, Wuhan, China. [7]RNA Institute, Wuhan University, Wuhan, China. [8]These authors contributed equally: Jing Wang, Weidong Liu. ✉E-mail: cjh@whu.edu.cn; haojian_zhang@whu.edu.cn

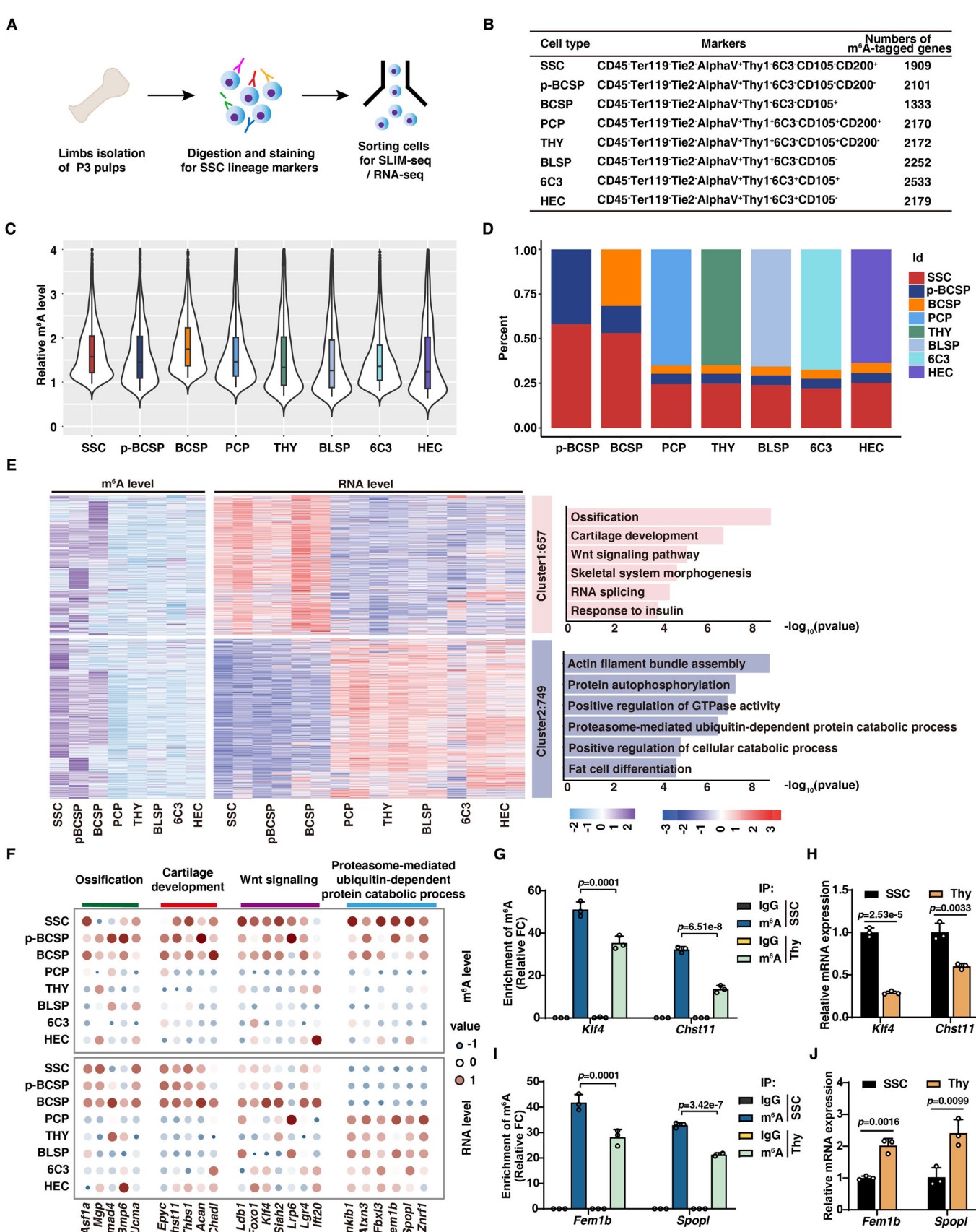

**Figure 1. Comprehensive m⁶A landscape across skeletal hierarchy.**

(A) Schematic diagram of isolation of mouse skeletal stem cells lineage for SLIM-seq and RNA-seq. (B) Numbers of m⁶A-tagged mRNA in different skeletal populations. (C) Violin plots showing relative m⁶A levels of different skeletal cells. $n = 2$, $n$ represents biological independent experiments. Data were presented as box-and-whisker plots (medians with interquartile range, and the whiskers extend to the smallest and largest data points within 1.5 times the IQR). (D) Bar plot showing the origin of m⁶A at each developmental stage from SSC to mature cells. (E) Comprehensive correlation and GO enrichment analysis of m⁶A and RNA levels in different populations. Heatmap showing 1406 m⁶A-tagged targets clustered with K-mean (K = 2), and clusters (C1-C2) were shown. Left is m⁶A and mRNA expression levels, right is GO enrichment analysis for cluster 1 and 2. Hypergeometric test was used for this statistical analysis. (F) Dot plot showing cell-type-specific m⁶A and mRNA expression levels for representative genes. The normalized levels of m⁶A and mRNA expression were centered on the mean of these eight populations, and both color and diameter of circles indicate the relative levels of m⁶A and mRNA expression. (G–J) MeRIP-PCR (G, I) and RT-qPCR (H, J) showing m⁶A and mRNA levels of indicated master regulators in SSCs and the Thy-subtype. $n = 3$, $n$ represents biological independent experiments. Data information: The data in (G, I) were analyzed by one-way ANOVA with Dunnett's multiple comparisons test. The data in (H, J) were analyzed by unpaired two-tailed Student's $t$ test. The error bars represent mean ± SD. Source data are available online for this figure.

remains elusive. Recently, we have successfully developed a highly sensitive and efficient super-low-input m⁶A sequencing (SLIM-seq) strategy to map the transcriptome-wide m⁶A tagged mRNAs in rare cells especially stem cells (Yin et al, 2021), which endows us the ability to explore the m⁶A landscape of skeletal stem cells.

Proteostasis is essential in maintaining stem cell function (Chua et al, 2023; Chua et al, 2020; Vilchez et al, 2012; Yadav et al, 2022). For instance, FEM1B is a substrate-recognition component of a Cul2-RING (CRL2) E3 ubiquitin-protein ligase complex. It recognizes C-degron of target proteins, which subsequently leads to their ubiquitination and degradation (Dankert et al, 2017; Koren et al, 2018; Manford et al, 2020). In this study, we decipher the landscape of RNA m⁶A methylome during skeletal hierarchy, and uncover the crucial roles of m⁶A in maintaining SSCs function via Mettl3-Fem1b-Gli1 mediated protein catabolism. This work establishes the link between epitranscriptomics and proteostasis in skeletal hierarchy, which provides key insights for bone development.

## Results

### Establishment of comprehensive RNA m⁶A landscape across skeletal hierarchy

We first portraited the m⁶A landscape of the skeletal system at the transcriptome-wide level. Following the previous strategy for mapping skeletal stem and progenitor cell populations (Chan et al, 2015; Liang et al, 2024), we isolated 8 distinct skeletal cellular populations from the epiphysis of postnatal day 3 (P3) mice using FACS, including SSC, pBCSP, BCSP, pro-chondrogenic progenitors (PCP), B-cell lymphocyte stromal progenitor (BLSP), the Thy subpopulation (Thy), the 6C3 subpopulation (6C3), and hepatic leukemia factor expressing cell (HEC) (Fig. 1A,B; Appendix Fig. S1A). By employing the SLIM-seq strategy developed previously by our group (Yin et al, 2021), we profiled m⁶A modification of these 8 distinct skeletal populations. To be noted, sorted cells from the same batch were equally separated and used for the corresponding RNA-seq, and the m⁶A profiles of all the biological replicates of a given cell type were computationally combined into meta-epitranscriptomic maps that provide a consensus m⁶A landscape as well as an initial assessment of variability within cell types.

With this dataset, we identified a total of 5327 m⁶A-tagged mRNAs, which were defined as high-confidence m⁶A targets in at least one of these 8 cell populations (Fig. 1B; Dataset EV1). To

validate the accuracy of these m⁶A profiles, we focused on skeletal stem and progenitor cells and profiled m⁶A modification upon Mettl3 deletion. Given that type II collagen alpha 1 (Col2a1) marks skeletal stem and progenitors in the growth plate (Cheng et al, 2023; Ono et al, 2014), we cross *Mettl3^{fl/fl}* with *Col2a1-Cre* mice to generate *Mettl3^{fl/fl};Col2a1-Cre* mice. For simplicity, *Mettl3^{fl/fl};Col2a1-Cre* mice will henceforth be referred to as *Mettl3^{KO}* mice and *Mettl3^{fl/fl}* control mice as wild-type (WT) mice. Deletion of Mettl3 was successfully achieved in *Mettl3^{KO}* mice, and dot blotting assay showed that Mettl3 loss also decreased m⁶A level in skeletal stem and progenitor cells (Appendix Fig. S1B–D). Consistently, compared with WT control, Mettl3 deficiency significantly decreased m⁶A levels in SSCs, Pre-BCSP and BCSP (Appendix Fig. S1E). Thus, these data verified the accuracy of m⁶A profiles.

We next explored the pattern of m⁶A modification in the skeletal hierarchy. Interestingly, m⁶A levels in skeletal stem and progenitors (SSC, Pre-BCSP, and BCSP) were higher than the levels in their downstream cell populations (PCP, THY, BLSP, HEC and 6C3) (Fig. 1C). We previously found that the establishment of cell fates in the hematopoietic hierarchy occurs at the root or early stage of lineage commitment, as the pattern of most m⁶A modifications in downstream cells mainly inherits from their upstream developmental stages(Yin et al, 2021). However, we found a different pattern of m⁶A modifications in the skeletal system. Similar to the hematopoietic system, the pattern of most m⁶A modifications (75%) in p-BCSP and BCSPs also inherited from SSCs, however, up to 65% m⁶A modifications in downstream cells (PCP, THY, BLSP, 6C3 and HEC) were de novo established (Fig. 1D). These data implied a sharp transition of cellular states during the differentiation of skeletal stem and progenitor cells into their downstream lineages (bone, cartilage and stroma). Therefore, these data indicate that RNA m⁶A plays a key role in cell fate determination in the skeletal hierarchy.

To further investigate the role of m⁶A in regulating cellular fate, we integrated and paired the m⁶A methylomes with gene expression profiles of these 8 populations. K-mean clustering of all 5327 m⁶A-tagged mRNAs revealed 9 major clusters (Appendix Fig. S1F). As exemplified, clusters 1 and 2 comprised m⁶A-tagged mRNAs that enriched in all three skeletal stem and progenitor populations (SSC, pBCSP and BCSP), cluster 3 contains BCSP-specific m⁶A-tagged mRNAs, and clusters 4-7 mainly enriched in downstream lineage cells. As expected, most of these m⁶A-tagged mRNAs (except targets in cluster 8 and 9) either positively or negatively correlated their mRNA levels. Next, we focused on cluster 1 and 2 as their higher m⁶A modifications in skeletal stem and progenitor populations (Fig. 1E). m⁶A-tagged mRNAs in

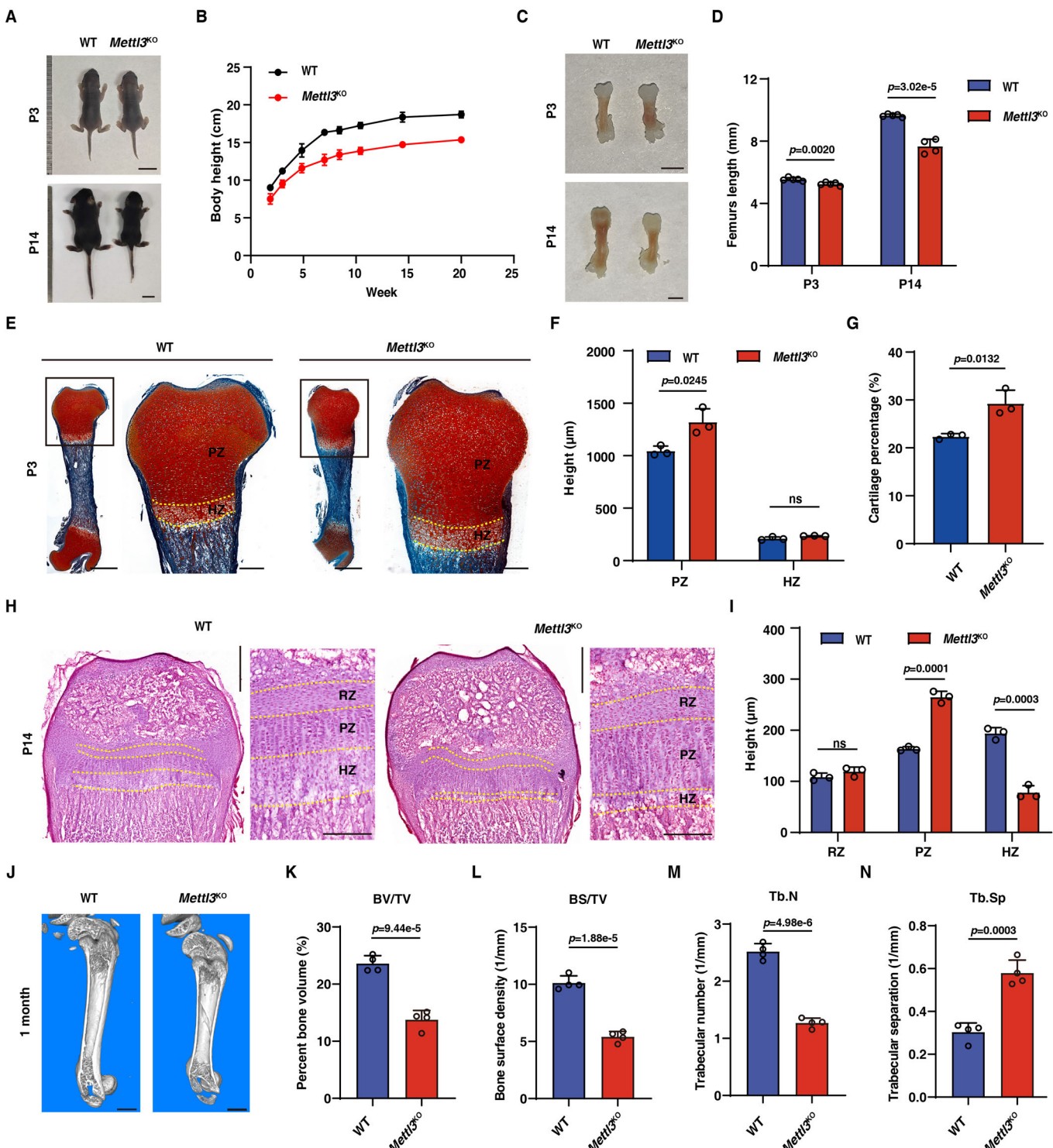

clusters 1 showed positive correlation with their mRNA levels, while a negative correlation was observed in cluster 2. GO analysis showed that the genes with higher m⁶A modifications and higher mRNA levels in cluster 1 were enriched in many important aspects of bone development, such as ossification, cartilage development, and Wnt signaling (Fig. 1F). Interestingly, the genes in cluster 2 showing higher m⁶A modifications and lower mRNA levels

mainly involved in actin filament bundle assembly, proteasome-mediated ubiquitin-dependent protein catabolic process (Fig. 1F). These findings were further validated by m⁶A-RIP-PCR and qRT-PCR. As exemplified, compared with the downstream Thy population, key regulators for stem cells (e.g., *Klf4*, *Chst11*, *Foxo1*, *Foxo3* and *Foxc1*) showed higher m⁶A levels with higher mRNA expression in SSCs (Fig. 1G,H; Appendix Fig. S1G,H); in contrast,

**Figure 2. Mettl3 deficiency causes bone dysplasia.**

(A) Gross observation of WT mice and *Mettl3*[KO] mice at P3 and P14. Scale bar, 1 cm. (B) Quantification of body height of WT mice and *Mettl3*[KO] mice at different ages (*n* = 5). Data are mean ± SD. (C) Gross observation of femurs of WT mice and *Mettl3*[KO] mice at P3 and P14. Scale bar, 2 mm. (D) Quantification of femur length of WT mice and *Mettl3*[KO] mice at P3 and P14 (*n* = 4 or 5). (E) Safranin O-fast green staining showing the cartilage area in femurs at P3. PZ, proliferation zone; HZ, hypertrophic zone. Scale bars, the left is 600 μm, the right is 200 μm. (F) Quantification of the heights of PZ and HZ (*n* = 3). (G) Quantification of the cartilage percentage at P3 (*n* = 3). (H) H&E staining showing the height of growth plates in femurs at P14. RZ, resting zone. Scale bars, the left is 500 μm, the right is 200 μm. (I) Quantification of the heights of RZ, PZ, and HZ (*n* = 3). (J) Micro-CT analysis showing the three-dimensional reconstruction images of cancellous bones of femurs at 1 month. Scale bar, 1 mm. (K–N) Quantification measurements of the bone volume/total volume (BV/TV), bone surface /total volume (BS/TV), trabecular number (Tb.N), and trabecular separation (Tb.Sp) of femurs from 1-month-old WT mice and *Mettl3*[KO] mice (*n* = 4) by micro-CT. Data information: The data in (D, F, G, I, K–N) were analyzed by unpaired two-tailed Student's *t* test. The error bars represent mean ± SD; ns, no significance. Source data are available online for this figure.

proteasome-related genes (e.g., *Fem1b*, *Spopl*) displayed higher m⁶A levels with lower mRNA levels in SSCs (Fig. 1I,J). Together, these data indicate a complicated and highly dynamic correlation between m⁶A and mRNA levels across the skeletal hierarchy.

## Mettl3 deficiency causes bone dysplasia

We next explored the role of m⁶A modifications in the skeletal hierarchy. Given the core role of METTL3 in catalyzing m⁶A modifications, we first assessed its expression in different skeletal populations. As expected, we found that Mettl3 expression exhibited highest level in SSCs, and gradually decreased along SSC differentiation (Appendix Fig. S2A). Higher expression of Mettl3 protein was also observed in the growth plate where skeletal stem and progenitors were enriched (Appendix Fig. S2B). Thus, these data drove us to dissect the function of Mettl3 in SSCs and skeletogenesis.

Interestingly, compared to WT control littermates, *Mettl3*[KO] mice displayed modest difference at postnatal day 3 (P3), but obvious dwarfism symptom as showing clearly the shorter body and less body weight at P14 (Fig. 2A,B; Appendix Fig. S2C). In particular, the femur in *Mettl3*[KO] mice was markedly shorter at P14 (Fig. 2C,D), indicating that Mettl3 deletion impairs skeletogenesis. We then examined the femoral growth plate on P3 and P14 by performing hematotoxylin and eosin (HE) staining, and saffron O-fast green staining. Interestingly, we observed an increased proliferating zone (PZ) at P3 in *Mettl3*[KO] mice, when compared to WT control (Fig. 2E–G; Appendix Fig. S2D). As development progresses, although the resting zone (RZ) were not significantly changed, the proliferating zone was obviously expanded with a concomitant decrease of the hypertrophic zone (HZ) in the metaphysis of *Mettl3*[KO] mice at P14 (Fig. 2H,I; Appendix Fig. S2E). TRAP staining showed that there was no significant difference in osteoclast activity between WT and *Mettl3*[KO] mice (Appendix Fig. S2F). Additionally, microCT (μCT) scanning of femurs showed that the ratios of bone volume v.s. tissue volume ratio (BV/TV) and bone surface v.s. tissue volume (BS/TV) were substantially reduced in *Mettl3*[KO] mice compared to WT mice at 1 and 3 month age (Fig. 2J–L; Appendix Fig. S2G,K,N,O). Mettl3 loss led to a decrease of the trabecular number (Tb.N) but an increase in the trabecular separation (Tb.Sp) (Fig. 2M,N; Appendix Fig. S2L,M). Interestingly, Mettl3 loss did not affect the bone cortical thickness (Appendix Fig. S2P,Q). Together, these results demonstrate that Mettl3 deletion impairs the development of skeletal stem progenitor cells and causes bone dysplasia.

## Mettl3 loss impairs the quiescence of SSC

We next investigated how Mettl3 influences SSC function (Fig. 3A). Unexpectedly, while pre-BCSP, BCSP and their downstream progenies

did not demonstrate significant changes, the frequency and total number of SSCs were obviously increased in *Mettl3*[KO] mice compared to WT control mice (Fig. 3B–D; Appendix Fig. S3A,B). Next, we assessed the cell cycle of SSCs, and found that the percentage of SSCs in G1 phase and cycling in *Mettl3*[KO] mice was clearly increased accompanied with an obvious decrease in G0 phase (Fig. 3E,F), indicating that Mettl3 deletion induced SSCs exiting quiescence and entering cell cycling. To examine the in vivo cell cycle status of SSCs, we performed EdU labeling by administering EdU into mice twice with 3-hour interval, and mice were euthanized at six hours. We observed more EdU labeling in the epiphyseal and growth plate area of femurs of *Mettl3*[KO] mice (Fig. 3G), indicating that more *Mettl3*[KO] SSCs are in the cycling phase. Consistently, Mettl3 deletion upregulated the expression of cell cycle related genes in SSCs including *Cdk4*, *Cdk6*, *Ccnd1*, *Ccne1*, and *Ccne2* (Appendix Fig. S3C). Collectively, these data demonstrate that Mettl3 is required for maintaining SSC quiescence.

## Loss of Mettl3 impairs the differentiation ability of SSCs

Further, we assessed whether Mettl3 influences the in vivo differentiation potential of SSCs. Sorted SSCs were transplanted beneath the renal capsules of healthy mice. Four weeks after transplantation, we performed μCT analysis, HE staining, and saffron O-fast green staining to analyze bone formation in vivo (Fig. 3H). As expected, SSCs from WT mice formed normal bone structure containing bone matrix and marrow-like cavity under renal capsule. However, SSCs from *Mettl3*[KO] mice displayed abnormal cartilage with aberrant accumulation of chondrocytes but lacking bone matrix and marrow-like cavity (Fig. 3I; Appendix Fig. S4A). Saffron O-fast green staining also confirmed this phenotype (Fig. 3J; Appendix Fig. S4B). Through analyzing by μCT, we confirmed that Mettl3 deletion impaired the bone formation capacity of SSCs (Fig. 3K,L). These data indicate that that Mettl3 deletion impairs the in vivo differentiation potential of SSCs. To further clarify this point, we conducted in vitro differentiation assay by culturing freshly sorted SSCs in osteogenic medium for 21 days (Appendix Fig. S4C). We found that *Mettl3*[KO] SSCs impaired osteogenic differentiation ability into mineralized nodules, as indicated by the marked decrease of alkaline phosphatase (ALP) staining and alizarin red S staining (Appendix Fig. S4D–F). In addition, compared to WT control, the expression levels of osteogenic genes (*Col1a1*, *Alpl*, *Ocn* and *Osterix*) were substantially down-regulated in *Mettl3*[KO] group (Appendix Fig. S4G). Consistently, knockdown of Mettl3 in SSCs by shRNA also damaged the differentiation ability into osteoblast, and caused the decreased expression of osteoblastic related genes (e.g. *Col1a1*, *Ocn*) (Appendix Fig. S4H,I). To rule out the off-target possibility, we restored Mettl3 expression in *Mettl3*[KO] SSCs. As expected, Mettl3 restoration rescued

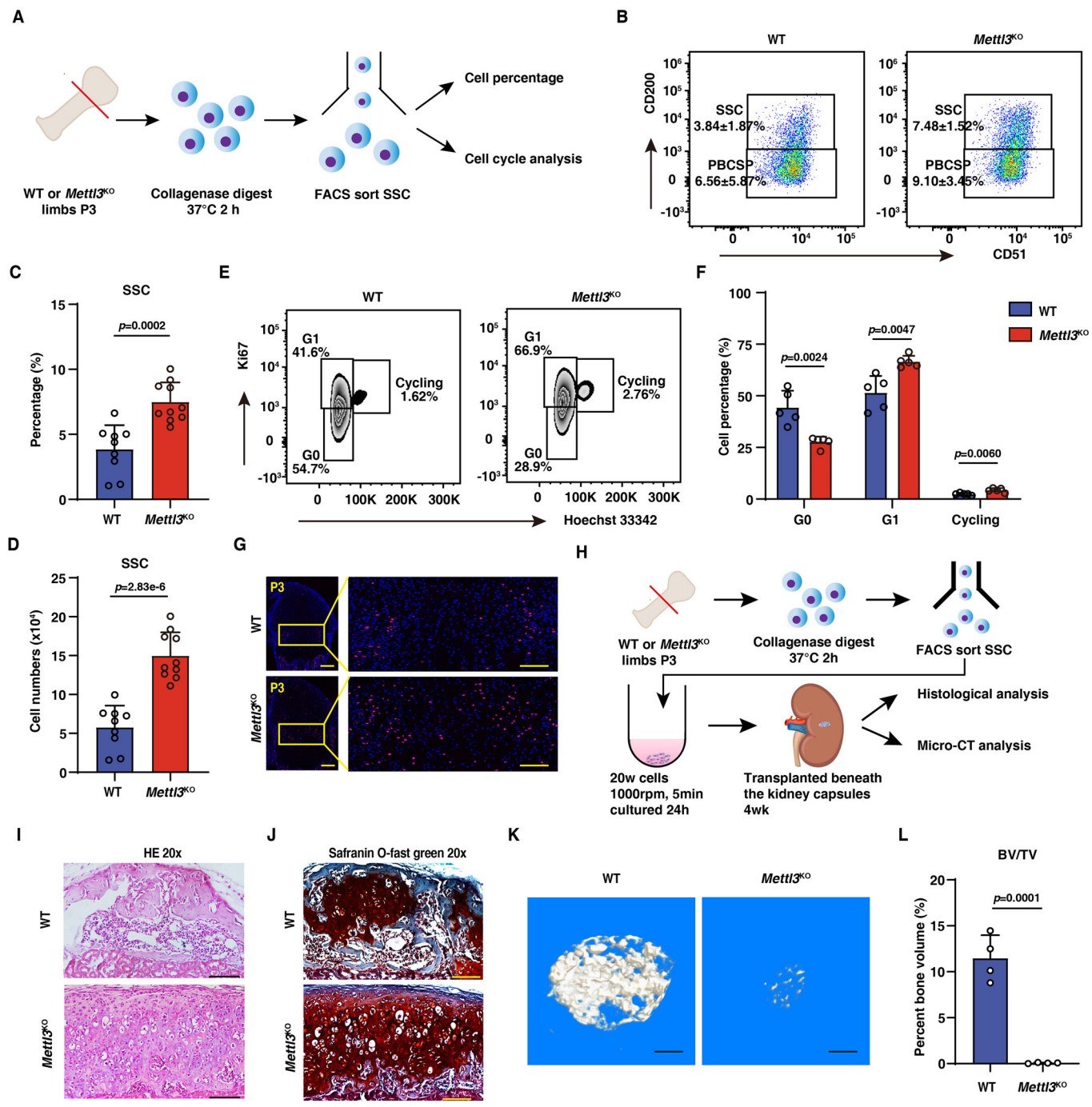

**Figure 3. Loss of Mettl3 impairs the quiescence and multipotency of SSCs.**

(A) Schematic diagram showing the analysis and sorting of SSCs lineages by flow cytometry. (B) Gated cell populations in flow cytometry of SSCs and PBCSPs in femurs of P3 WT mice and *Mettl3*KO mice. (C) Quantification of SSCs population percentages ($n = 9$ or 10). (D) Quantification of SSCs population cell numbers ($n = 9$ or 10). (E) Flow cytometry analysis of cell cycle phase in SSCs of P3 from WT mice and *Mettl3*KO mice. S and G2-M were combined and defined as cycling. (F) Cell cycle analysis showing percentages of cell cycle phases in SSCs of P3 from WT mice and *Mettl3*KO mice ($n = 5$). (G) EdU labeling assay of distal-femur growth plates (pulsed on P3). Scale bars, the left is 200 μm, the right is 100 μm ($n = 3$). (H) Schematic diagram showing renal capsule transplantation analysis of SSCs in vivo. (I) H&E staining of tissue grafts following cell transplant beneath the renal capsule. The uncropped image was displayed in Appendix Fig. S4A. Scale bar, 100 μm. (J) Safranin O-fast green staining of tissue grafts following cell transplant beneath the renal capsule. The uncropped image was displayed in Appendix Fig. S4B. Scale bar, 100 μm. (K) Micro-CT analysis showing the three-dimensional reconstruction images of tissue grafts at 1 month. Scale bar, 200 μm. (L) Quantification measurements of BV/TV of tissue grafts by micro-CT, SSCs from WT mice and *Mettl3*KO mice ($n = 4$). Data information: The data in (C, D, F, L) were analyzed by unpaired two-tailed Student's *t* test. The error bars represent mean ± SD. Source data are available online for this figure.

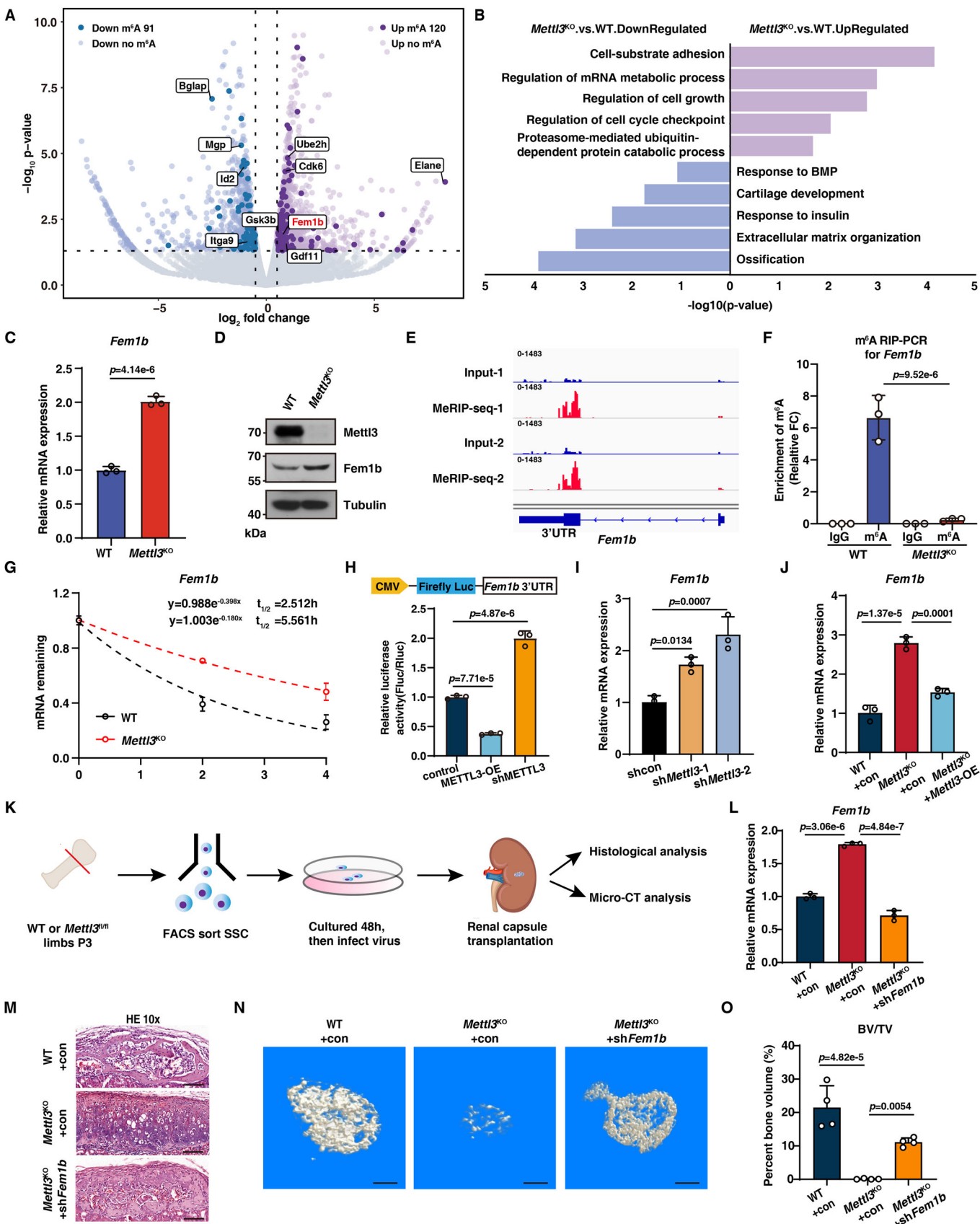

**Figure 4. Mettl3 regulates Fem1b stability to maintain SSCs function.**

(A) Volcano plots of DEGs in *Mettl3*[KO] compared with WT SSCs. m[6]A-tagged DEGs were highlighted with different colors. Negative Binomial Regression Analysis was used for data analysis. *n* = 2, *n* represents biological independent experiments. (B) GO analysis showing terms enriched in differential pathways for upregulated or downregulated m[6]A-tagged DEGs in *Mettl3*[KO] SSCs compared with WT SSCs. Hypergeometric test was used for data analysis. (C) qRT-PCR analysis showing *Fem1b* mRNA expression in WT SSCs and *Mettl3*[KO] SSCs. *n* = 3, *n* represents biological independent experiments. (D) Western blot analysis showing Fem1b protein expression in WT SSCs and *Mettl3*[KO] SSCs. (E) IGV tracks showing distribution m[6]A peaks of *Fem1b* transcripts in mesenchymal stem cells. (F) MeRIP-PCR analysis of m[6]A enrichment of mRNAs for *Fem1b* in WT SSCs and *Mettl3*[KO] SSCs. *n* = 3, *n* represents biological independent experiments. (G) *Fem1b* mRNA half-life of in WT SSCs and *Mettl3*[KO] SSCs. *n* = 3, *n* represents biological independent experiments. (H) Relative luciferase activity of FEM1B 3'UTR in HEK293T cells transfected with Control, METTL3-OE, shMETTL3. *n* = 3, *n* represents biological independent experiments. (I) qRT-PCR analysis of the expression of *Fem1b* after knocking down Mettl3. *n* = 3, *n* represents biological independent experiments. (J) qRT-PCR analysis of the expression of *Fem1b* after overexpressing Mettl3. *n* = 3, *n* represents biological independent experiments. (K) Schematic diagram showing the renal capsule transplantation rescue Mettl3-deficient of SSCs in vivo. (L) qRT-PCR analysis of *Fem1b* mRNA expression in WT SSCs and *Mettl3*[KO] SSCs transduced with shcontrol or sh*Fem1b*. *n* = 3, *n* represents biological independent experiments. (M) H&E staining of tissue grafts following cell transplant beneath the renal capsule. The uncropped image was displayed in Appendix Fig. S5D. Scale bar, 100 μm. (N) Micro-CT analysis showing the three-dimensional reconstruction images of tissue grafts at 1 month. Scale bar, 200 μm. (O) Quantification measurements of BV/TV of tissue grafts by micro-CT, SSCs from WT mice and *Mettl3*[KO] mice transduced with shcontrol or sh*Fem1b* (*n* = 4). Data information: The data in (C) was analyzed by unpaired two-tailed Student's *t* test. The data in (F, H–J, L, O) were analyzed by one-way ANOVA with Dunnett's multiple comparisons test. The error bars represent mean ± SD. Source data are available online for this figure.

the osteogenesis deficiency and reversed the expression of osteoblastic related genes (e.g. *Col1a1*, *Ocn*) in *Mettl3*[KO] SSCs (Appendix Fig. S4J,K). Collectively, these data demonstrated that Mettl3 is essential for maintaining SSC quiescence and multipotency.

## Mettl3 regulates Fem1b stability to maintain SSCs function

To understand the mechanisms of how Mettl3 sustains SSCs function, we performed RNA-seq and SLIM-seq to globally analyze gene expression in *Mettl3*[KO] SSCs. When compared to WT SSCs, a total of 211 differentially expressed genes (DEGs) with m[6]A modification were altered in *Mettl3*[KO] SSCs, including 120 upregulated (e.g. *Ube2h*, *Cdk6*, *Fem1b*) and 91 downregulated (e.g. *Bglap*, *Mgp*, *Id2*) respectively, suggesting these genes might be regulated by Mettl3 in a m[6]A-dependent manner (Fig. 4A; Dataset EV1). Gene Ontology (GO) analysis showed that the downregulated m[6]A-tagged DEGs were specifically enriched in the biological processes including ossification, extracellular matrix organization, and cartilage development. BMP signaling pathway related genes (e.g. *Bmpr1a* and *Bglap*) were also among these downregulated m[6]A-tagged DEGs. We confirmed that their mRNA and m[6]A levels were decreased after Mettl3 deletion in SSCs (Appendix Fig. S5A,B), suggesting that the BMP signaling pathway is regulated by RNA m[6]A modification in SSCs. The upregulated m[6]A-tagged DEGs were related to regulation of mRNA metabolic process, cell growth, regulation of cell cycle checkpoint, and proteasome-mediated ubiquitin-dependent protein catabolic process (Fig. 4B).

Proteostasis adjusts protein composition and maintains protein integrity, and has emerged as fundamentally and preferentially important for stem cells (Llamas et al, 2020). Skeletal stem cells reside in an extremely nutrient-deficient niche, which requires adapting the anabolic and catabolic system, especially proteostasis, for proper cell fate determination. However, the demand-adapted regulatory circuits of SSCs remain unknown. Interestingly, we found that the upregulated m[6]A-tagged DEGs in *Mettl3*[KO] SSCs were enriched in the proteasome-mediated ubiquitin-dependent protein catabolic process (Fig. 4B). Among these potential candidates (e.g. Fem1b, Ube2h), Fem1b is the substrate-recognition component of a Cul2-RING (CRL2) E3 ubiquitin-protein ligase complex. Thus, we focused on Fem1b, and found that Fem1b expression was obviously upregulated in *Mettl3*[KO] SSCs

(Fig. 4C,D). By analyzing the publicly available MeRIP-seq data (Wu et al, 2018), we observed a significant m[6]A peak in the 3'UTR of Fem1b (Fig. 4E). Next, focusing on this 3'UTR region, we performed m[6]A RIP-PCR and found that m[6]A modification was significant enriched in this region, which was nearly abrogated in *Mettl3*[KO] SSCs (Fig. 4F). Meanwhile, Mettl3 deletion increased *Fem1b* mRNA stability in SSCs (Fig. 4G), indicating that m[6]A modification promotes the decay of *Fem1b* mRNA. To further examine the effect of m[6]A modification on *Fem1b* mRNA stability, we selected this m[6]A-enriched *Fem1b* 3'UTR region ( ~ 450 bp) to construct the firefly luciferase reporter, and the m[6]A level of Fem1b 3'UTR in this reporter was significantly increased by Mettl3 overexpression, which validated the accuracy of this reporter (Fig. 4H; Appendix Fig. S5C). As expected, we found that Mettl3 overexpression reduced the luciferase activity; by contrast, knock-down of Mettl3 increased the luciferase activity (Fig. 4H). Similarly, we found that *Mettl3* knockdown by lentivirus infection markedly upregulated *Fem1b* mRNA level (Fig. 4I). The enhanced expression of Fem1b in *Mettl3*[KO] SSCs was markedly rescued by restoration of Mettl3 (Fig. 4J). Thus, these data suggest that Mettl3 suppresses Fem1b expression in a m[6]A-dependent manner.

We further assessed whether Fem1b mediates the function of Mettl3 in SSC. We then knocked down Fem1b expression in *Mettl3*[KO] SSCs, and performed the renal capsule transplantation and osteogenesis phenotype (Fig. 4K,L). As expected, we found that Fem1b knockdown partially rescued the impaired osteogenic differentiation ability of *Mettl3*[KO] SSCs, as determined by the appearance of bone matrix and the increased calcified nodules (Fig. 4M; Appendix Fig. S5D–F). Meanwhile, μCT analysis also showed that Fem1b knockdown markedly rescued the in vivo bone formation ability of *Mettl3*[KO] SSCs (Fig. 4N,O). This phenotype was in line with the increased expression of the osteogenic genes, such as *Col1a1*, *Alpl*, and *Ocn* (Appendix Fig. S5G). Together, these data demonstrate that Fem1b acts as a functional downstream target of Mettl3 in SSCs.

## Gli1 mediates the function of Mettl3 in maintaining the function of SSCs

The above findings prompted us to study how Fem1b regulates SSCs function. To identify the potential substrates of Fem1b, we expressed Flag-tagged Fem1b (Flag-Fem1b) in both 293T cells and skeletal cells,

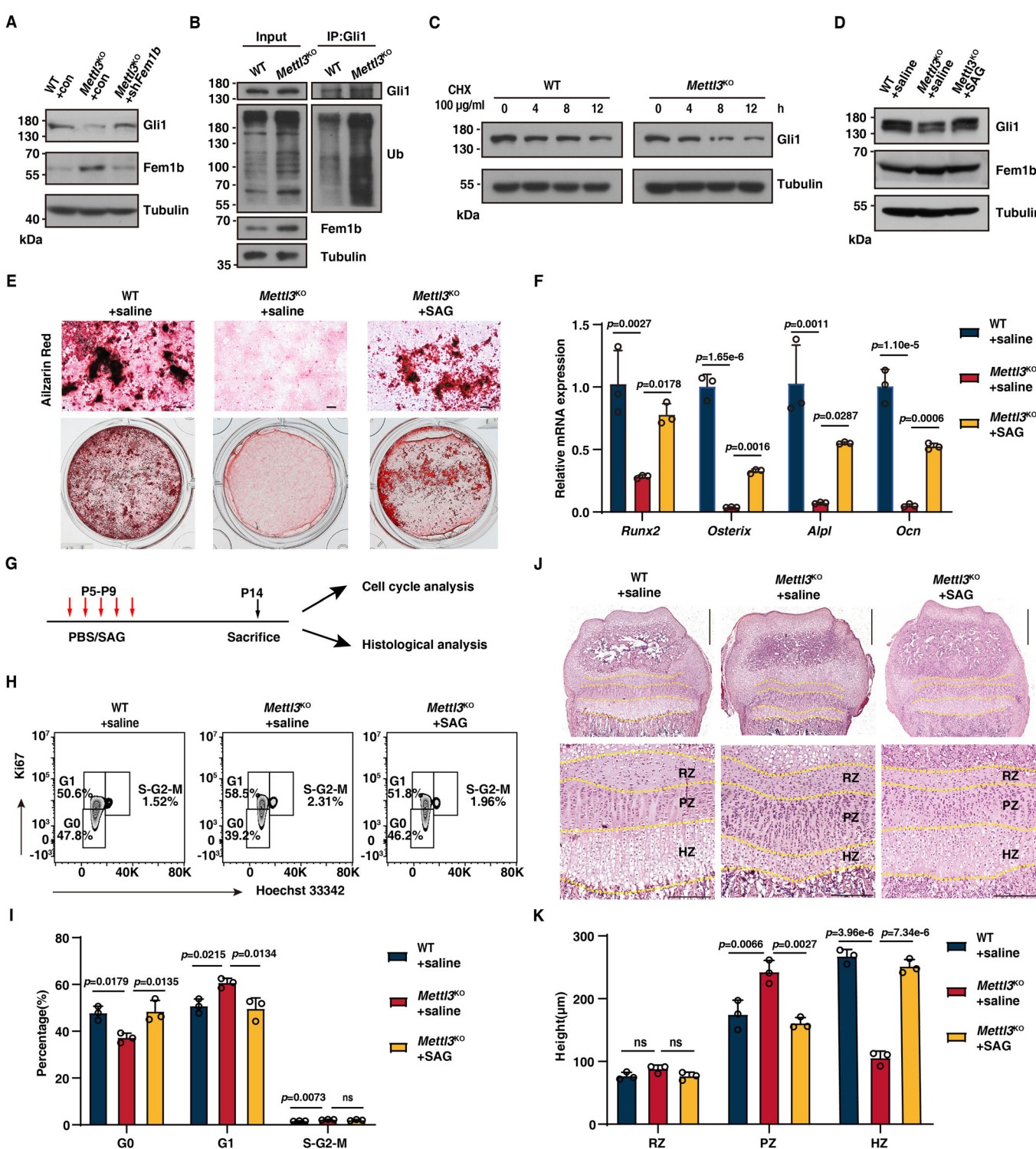

and performed liquid chromatograph-mass spectrometry (LC-MS) analysis (Appendix Fig. S6A). A total of 127 and 96 proteins in skeletal cells and 293T cells respectively were identified by LC-MS analysis. Integrating these two datasets, we identified 16 overlapped proteins (e.g. Col1a1, Gli1, Myh9, Pdik1l), which served as potential substrates of Fem1b in both skeletal cells and 293T (Appendix Fig. S6B). Interestingly, among these potential substrates, Gli1 was one of the top

candidates (Appendix Fig. S6C). Gli1 acts as a transcriptional factor under the downstream of Hedgehog signaling, and is responsible for bone formation and fracture repair (Jeffery et al, 2022; Shi et al, 2017). Thus, we focused on Gli1 and conducted co-IP assay. As expected, we confirmed the interaction between Fem1b and Gli1 (Appendix Fig. S6D). Therefore, these data prompted us to focus on Gli1 as a downstream substrate of Fem1b. To be noted, we did not observe

**Figure 5. Gli1 mediates function of Mettl3 in maintaining the function of SSCs.**

(A) Western blot assay showing Fem1b and Gli1 expression in WT SSCs and *Mettl3*<sup>KO</sup> SSCs transduced with shcontrol or sh*Fem1b*. Actin served as the loading control. (B) WT SSCs and *Mettl3*<sup>KO</sup> SSCs were harvested for endogenous ubiquitination assays. Cell lysates were immunoprecipitated with Anti-Gli1 antibody and then analyzed by immunoblotting using the indicated antibodies. (C) WT SSCs and *Mettl3*<sup>KO</sup> SSCs were treated with 100 µg/mL CHX and harvested at the indicated time followed by Western blotting analysis. (D) Western blot assay showing rescue of Gli1 expression in WT SSCs and *Mettl3*<sup>KO</sup> SSCs treated with saline or SAG. (E) Alizarin red staining of osteogenic differentiation for 21 d of WT SSCs and *Mettl3*<sup>KO</sup> SSCs treated with saline or SAG. Representative images are shown ($n = 3$ per genotype). Scale bar, 200 µm. (F) qRT-PCR analysis of mRNA expression levels of osteogenic genes in WT SSCs and *Mettl3*<sup>KO</sup> SSCs treated with saline or SAG. $n = 3$, $n$ represents biological independent experiments. (G) Experimental scheme for SAG injection assay to analyze the functional rescue in vivo. (H) Flow cytometry analysis of cell cycle phase of SSCs from WT mice and *Mettl3*<sup>KO</sup> mice administrated with or without SAG ($n = 3$). (I) Quantification of each cell cycle phase of SSCs from WT mice and *Mettl3*<sup>KO</sup> mice administrated with or without SAG ($n = 3$). (J) H&E staining of growth plates in WT mice and *Mettl3*<sup>KO</sup> mice administrated with or without SAG. Scale bars, upper is 500 µm, lower is 200 µm. (K) Quantification of the height of RZ, PZ, and HZ of growth plates in WT mice and *Mettl3*<sup>KO</sup> mice administrated with or without SAG ($n = 3$). Data information: The data in (F, I, K) was analyzed by one-way ANOVA with Dunnett's multiple comparisons test. The error bars represent mean ± SD; ns, no significance. Source data are available online for this figure.

obvious m⁶A peak on the *Gli1* mRNA, and there was also no difference in the expression level of *Gli1* mRNA in SSCs from WT and *Mettl3*<sup>KO</sup> mice (Appendix Fig. S6E,F), suggesting that Gli1 is not a direct m⁶A target. Intriguingly, we observed an obvious decrease of Gli1 protein in *Mettl3*<sup>KO</sup> SSCs, which was rescued upon restoration of Fem1b (Fig. 5A). Previous evidence showed that Fem1b could interact with Gli1 and promote its ubiquitylation (Gilder et al, 2013). In here, we observed an obvious increase of the polyubiquitination level of Gli1 in *Mettl3*<sup>KO</sup> SSCs (Fig. 5B). This increased ubiquitination accelerated Gli1 decay in *Mettl3*<sup>KO</sup> SSCs, as showing by cycloheximide (CHX) chase assays (Fig. 5C). To further examine whether the ubiquitination of Gli1 is dependent on Fem1b, we knocked down Fem1b in SSCs, and found that Fem1b deletion decreased Gli1 ubiquitination accompanying with an increased expression of Gli1 protein (Appendix Fig. S6G,H). We also assessed whether Fem1b depletion reverses the increased ubiquitination and degradation of Gli1 in *Mettl3*<sup>KO</sup> cells. As expected, the ubiquitination level and degradation of Gli1 was obviously reversed in *Mettl3*<sup>KO</sup> cells upon Fem1b loss (Appendix Fig. S6I,J). Together, these results reveal an epitranscriptomic program that promotes GLI1 protein stability via METTL3-FEM1B axis in SSCs.

Next, we assessed whether Gli1 mediates the function of Mettl3 in SSCs. We first overexpressed Gli1 in *Mettl3*<sup>KO</sup> SSCs and performed osteogenesis in vitro. We found that Gli1 overexpression rescued the defective osteogenic differentiation ability of *Mettl3*<sup>KO</sup> SSCs (Appendix Fig. S6K,L). Next, we applied an agonist, SAG, to activate the Hedgehog pathway. As expected, SAG treatment increased *Gli1* transcription in a dose-dependent manner but not *Fem1b* expression (Appendix Fig. S6M,N). Interestingly, SAG treatment partially rescued osteogenesis capability of *Mettl3*<sup>KO</sup> SSCs in vitro, and also reversed the expression of osteogenic genes (e.g. *Runx2, Osteorix, Alpl, Ocn*) (Fig. 5D–F). Moreover, we treated WT and *Mettl3*<sup>KO</sup> mice at the neonatal stage with SAG. Cell cycle analysis showed that SAG treatment partially restored the quiescent state of SSCs quiescent (Fig. 5G–I). Meanwhile, histological analysis also showed that injection of SAG restored the PZ and HZ zone in the growth plate of *Mettl3*<sup>KO</sup> mice to the similar level of WT mice (Fig. 5J,K). Together, our data indicate that Gli1 mediates the function of Mettl3 in maintaining the function of SSCs.

## Discussion

Understanding the regulatory network of skeletal hierarchy is a fundamental scientific question in the field. Here, we portrait a comprehensive m⁶A landscape along the developmental trajectory of skeletal stem progenitors, and uncover a new epitranscriptomic program, METTL3-FEM1B-GLI1 axis, that controls skeletal stem cell quiescence and potency. These findings provide a key insight into the development of skeletal system, and also establish a foundation for exploring the molecular mechanisms of bone pathophysiology.

To our knowledge, we depicted the first m⁶A landscape of the skeletal hierarchy. Bone development is regulated by various signaling pathways, and recent studies have unveiled the roles of some epitranscriptomic mechanisms in skeletal differentiation and specific cell lineage development (Gu et al, 2024; Wu et al, 2018; Yao et al, 2019). In here, this study establishes a direct insight across the skeletal system. Surprisingly, we found that skeletal stem progenitors have higher m⁶A levels that those in the downstream populations, and uncovered a complicated and highly dynamic correlation between m⁶A and mRNA levels across the skeletal hierarchy. It should be noted that SSCs is defined using the immunophenotype markers as previous study (Chan et al, 2015; Liang et al, 2024). Although we cannot exclude the contamination of other osteolineage cells in SSC population, we do not believe this issue could affect the whole m⁶A landscape of the skeletal hierarchy we observed. m⁶A modification plays a decisive role in cell identity transformation. For instance, accumulating studies indicate Mettl3 plays important roles in the regulation of mesenchymal stem cell (MSC) fate, dental pulp stem cells (DPSCs) differentiation, and hematopoietic stem cell (HSC) homeostasis and niche generation (Cai et al, 2021; Gao et al, 2024; Wu et al, 2018; Yin et al, 2021). However, the relationship between epigenetic regulation and maintenance of SSC function remains unclear. Here, we elucidated the function of Mettl3 in early bone development and the molecular mechanism of Mettl3 in maintaining the stemness of SSC. It is also notable that deletion of Mettl3 in different populations and stages of osteoblasts cells mediated by different Cre system (e.g. *Prx1-Cre, Ocn-Cre, Lepr-Cre*) could cause distinct but overlapping phenotypes in bone (Gao et al, 2024; Wu et al, 2018). In addition, previous fate-mapping studies showed that cells expressing *Col2-cre* recombinase contribute to multiple cell types including chondrocytes, perichondrial precursors, osteoblasts, and stromal cells in the skeletal system (Ono et al, 2014). Thus, it is reasonable that Mettl3 deletion in these different cells may result in compensatory phenotypes. This might explain our observation that the bone cortical thickness did not change upon Mettl3 deficiency. Overall, combining with these works, our study further reflects the high

dynamics of epitranscriptomic programs in regulating skeletal development and cell fate determination.

Our study reveals a novel mechanism for controlling proteostasis of SSCs. Increasing evidence demonstrate that protein homeostasis is essential for stem cell maintenance, and disruption of proteostasis normally impairs stem cell self-renewal and is associated with cancer predisposition syndromes, degenerative disorders and age-related pathologies in vivo (Chua et al, 2023; Lv et al, 2021; Vilchez et al, 2014). Given the hypoxic, and extremely nutrient-deficient microenvironment that skeletal stem cells locate, it requires SSCs to adapt the anabolic and catabolic system to meet its demand for proper cell fate maintenance. Intriguingly, we found that Mettl3-mediated $m^6A$ modification regulates *Fem1b* mRNA stability, the key substrate-recognition subunit of CRL2 E3 ubiquitin-protein ligase complex, which subsequently influences proteostasis of SSCs. As exemplified, Gli1 is one of the Fem1b substrates in SSCs. Gli1 is also an important transcriptional regulator downstream of Hedgehog pathway, contributing to regulate bone development (Hu et al, 2023). Given that Gli1 is relatively widely expressed in different skeletal cell types, it is unknow whether other types of skeletal cells contribute to SSC maintenance which need to be investigated in the future. Our work establishes the link between epitranscriptome and proteostasis that is mediated by Mettl3-Fem1b axis, and unveils a regulatory mechanism of how SSCs maintain its proteostasis.

In summary, this study establishes a previously unexplored landscape of $m^6A$ modification, and uncovers a crucial role of the epitranscriptomic program in maintaining SSCs quiescence and potency by regulating proteostasis via Mettl3-Fem1b-Gli1 axis. This study sheds a keen light on bone development.

# Methods

### Reagents and tools table

| Reagent/resource | Reference or source | Identifier or catalog number |
|---|---|---|
| **Experimental models** | | |
| HEK293T | ATCC | Cat#CRL-3216 |
| Mouse: *Mettl3*fl/fl | Biocytogen | N/A |
| Mouse: Col2-cre | Biocytogen | N/A |
| Mouse: C57BL/6J | Jackson Laboratory | Cat#000664 |
| **Recombinant DNA** | | |
| pLKO.1 | Addgene | Cat#8453 |
| pLKO.1-sh*Mettl3* | This study | N/A |
| pMIR-REPORT | ThermoFisher | Cat#AM5795 |
| pMIR-REPORT-*FEM1B* | This study | N/A |
| MSCV-KD-G2P | This study | N/A |
| MSCV-*Mettl3*-G2P | This study | N/A |
| MSCVnm-Puro-IRES-GFP-Intron | This study | N/A |
| MSCVnm-sh*Fem1b*-G2P | This study | N/A |
| **Antibodies** | | |
| $m^6A$ | Abcam | Cat#ab151230 |

| Reagent/resource | Reference or source | Identifier or catalog number |
|---|---|---|
| METTL3 | Proteintech | Cat#15073-1-AP |
| FEM1B | Proteintech | Cat#11030-1-AP |
| GLI1 | Proteintech | Cat#66905-1-Ig |
| Ub | Cell Signaling Technology | Cat#3936s |
| Tubulin | Abcam | Cat#ab6160 |
| IgG | Proteintech | Cat#B900610 |
| Biotin anti-mouse CD45 | Biolegend | Cat#103104 |
| Biotin anti-mouse TER-119 | Biolegend | Cat#116204 |
| Biotin anti-mouse CD202b (Tie-2) | Biolegend | Cat#124006 |
| PerCP/Cyanin5.5 Streptavidin | Biolegend | Cat#405214 |
| Anti-mouse CD51-PE | Biolegend | Cat#104106 |
| Anti-mouse 6C3-PE-Cy7 | Biolegend | Cat#108314 |
| Anti-mouse Thy1.1-APC-Cy7 | Biolegend | Cat#202520 |
| Anti-mouse Thy1.2-APC-Cy7 | Biolegend | Cat#105328 |
| Anti-mouse CD105-FITC | Biolegend | Cat#120406 |
| Anti-mouse CD200-APC | Biolegend | Cat#123809 |
| Anti-mouse Ki67-PE-CF594 | Biolegend | Cat#652427 |
| Hoechst 33342 | ThermoFisher | Cat#H3570 |
| 7AAD | Stem Cell | Cat#75001 |
| **Oligonucleotides and other sequence-based reagents** | | |
| Primers for RT-PCR, see Appendix Table S1 | This study | N/A |
| Primers for mouse genotyping, see Appendix Table S1 | This study | N/A |
| Primers for shRNA, see Appendix Table S1 | This study | N/A |
| Fem1b 3'UTR sequence for luciferase assay | This study | N/A |
| **Chemicals, enzymes and other reagents** | | |
| Polybrene | Sigma-Aldrich | Cat#H9268 |
| Actinomycin D | Sigma-Aldrich | Cat#A9415 |
| Cycloheximide(CHX) | MCE | Cat#HY-12320 |
| Smoothened Agonist(SAG) | Selleck | Cat#S7779 |
| Collagenase | Sigma-Aldrich | Cat#6885 |
| Protein-A beads | ThermoFisher | Cat#10002D |
| Library Preparation VATHS mRNA Capture Beads | Vazyme | Cat#N401-01 |
| TruSeq Stranded mRNA Library Prep Kit | Illumine | Cat#RS-122-2101 |
| Pure Beads Kit | KAPA | Cat#KK8002 |
| KAPA Hifi HS RM | KAPA | Cat#KK2602 |
| ReverTra Ace qPCR RT Kit | TOYOBO | Cat#FSQ-101 |
| Dual-Luciferase Reporter Assay System | Promega | Cat#1980 |
| Hematoxylin and Eosin Staining Kit | Beyotime | Cat#C0105S |
| Safranin O- fast green cartilage staining kit | Solarbio | Cat#G1371 |

| Reagent/resource | Reference or source | Identifier or catalog number |
|---|---|---|
| Alcian Blue Cartilage Stain solution | Servicebio | Cat#G1027 |
| Alizarin red S | Sigma-Aldrich | Cat#A5533 |
| **Software** | | |
| GraphPad Prism 8 | GraphPad Software | https://www.graphpad.com/ |
| FlowJo software (version 10.8.1) | FlowJo | https://www.flowjo.com/ |
| **Other** | | |
| RNA-seq | This study | GEO: GSE284224 |
| SLIM-seq | This study | GEO: GSE284224 |

## Mice

C57BL/6J (CD45.2) background *Mettl3* mice and *Col2a1-Cre* mice were obtained from Biocytogen, China. *Mettl3^{fl/fl}* mice were crossed with *Col2a1-Cre* mice. Mouse femurs samples on postnatal day (P) 3 were collected from at least three or more independent litters regardless of sex, while femurs samples on P14 and 1 month and 3 month were collected from male mice. Randomization was performed among mice of the same genotype. All mice were bred and maintained in Animal Center of Medical Research Institute at Wuhan University. All animal experiments were performed according to the protocols approved by the Animal Care and Use Committee of the Medical Research Institute, Wuhan University (201806).

## Histological analysis

The bone tissues were dissected, fixed in 4% paraformaldehyde (PFA) at a sufficient volume for 24 h at 4 °C. Samples were decalcified in 10% ethylenediaminetetraacetic acid (EDTA) for 14 d on a shaker at room temperature. Then, the tissues were dehydrated, embedded in paraffin and sectioned at 5 μm for hematoxylin and eosin (H&E), Safranin O (Safo) and fast green staining, and immunohistochemistry. For H&E staining, the sections were stained with hematoxylin for 10 s, running water for 1 min, eosin for 10 s and running water for 1 min. For Safo and fast green staining, the sections were stained with 0.2% fast green for 5 min, washed, stained with 0.1% Safranin O, and washed with 1% glacial acetic acid. Finally, slides were mounted in neutral mounting medium.

## Micro-computed tomography (micro-CT)

Mouse femurs of male mice at 1 month and 3 month were harvested, fixed overnight in 4% paraformaldehyde, and scanned using the Scanco 50 micro-CT system (Switzerland). Projection data were reconstructed using the NRecon software and the data were analyzed using the CT Analyzer software. The specimens were scanned at the shooting parameters (resolution of 10 μm, a voltage of 50 kV, and a 0.5-mm aluminum filter). Trabecular bone data were manually set at 200 layers below the growth plate at a region of interest (ROI), for quantification using BV/TV, BS/TV, Tb.N, and Tb.Sp.

## Immunohistochemical staining

Paraffin sections of mouse femurs were de-waxed and rehydrated with graded alcohol. Then antigen repair was performed with pepsin for 30 min at 37 °C. Peroxidase blocking with 3% $H_2O_2$ for 25 min at room temperature and goat serum blocking for 30 min at 37 °C was performed. A primary antibody was added and incubated overnight at 4 °C. The secondary antibody (anti-mouse or anti-rabbit) was added and incubated for 1 h at 37 °C. Washing with PBS after incubation at each step. The sections were treated with horseradish peroxidase for 40 min at 37 °C and stained with DAB under microscope observation.

## Mice treated with Hedgehog agonist

For the Hedgehog agonist experiment in vivo, WT mice and *Mettl3^{KO}* mice on P5 were randomly divided into three groups: WT-saline group, *Mettl3^{KO}*-saline group and *Mettl3^{KO}*-SAG group. Smoothened Agonist (SAG) (25 mg/kg, Selleck, USA) was dissolved in saline. Saline or SAG were administered intraperitoneally beginning on P5 and continuing for 5 days. The femurs of the P14 mice were then used for histological and cell cycle analysis.

## Isolation and sorting SSCs of primary murine

Isolation and identification of mouse SSCs were performed according to methods published in previous studies with reasonable minor modifications (Chan et al, 2015; Gulati et al, 2018). In brief, lower limbs were collected from mice (regardless of sex) on P3, and metaphyses lacking the bone marrow and periosteum were dissociated. Tissue fragments, including the growth plates, were mechanically digested and then digested with 0.2% collagenase digestion solution for 1 h at 37 °C under shaking at 220 rpm. The cell suspension was filtered through 100-μm nylon mesh and then precipitated for 5 min to wipe off collagen fiber.

The cell suspension was collected for staining with antibodies against CD45-biotin (103104, BioLegend, USA), Ter119-biotin (116204, BioLegend), and Tie2-biotin (124006, BioLegend) for 20 min on ice and washed with PBS; then stained with fluorochrome-conjugated antibodies of PerCP/Cyanine5.5 Streptavidin (405214, BioLegend), CD51-PE (104106, BioLegend), Thy1.1-APC/Cyanine7 (202520, BioLegend), Thy1.2-APC/Cyanine7 (105328, BioLegend), 6C3-PE/Cyanine7 (108314, BioLegend), CD105-FITC (120406, BioLegend), and CD200-APC (123809, BioLegend) for 20 min on ice. The viability dye 7-amino actinomycin D (7-AAD, 75001, StemCell Technologies, USA) was used to exclude dead cells. Gates were selected as determined by internal fluorescence-minusone controls to separate positively and negatively stained cell populations according to a previously published method (Chan et al, 2018). The cells were sorted by BD FACSAria III, USA. Obtained raw data were further analyzed using the FlowJo software (v.10.1). Freshly sorted SSCs, pBCSP, BCSP and the other subgroups were subjected to populational and functional assays or RNA-seq and SLIM-seq analyses.

## Cell cycle

To determine cell cycle distribution and quiescence of SSCs, digested cells were cultured with Hoechst 33342 (Life Technologies) for 90 min at 37 °C and then stained as described previously. The cells were then fixed with Paraformaldehyde and stained with PE-CF594-anti-Ki67 in PermWash solution for 30 min at 4 °C. Cells were washed once and resuspended in PermWash solution and analyzed by flow cytometry. Data was analyzed using FlowJo software.

## EdU labeling assay

EdU dissolved in PBS was administered to pups twice, at 6 and 3 h before euthanization at the indicated postnatal days (50 μg for P3 per injection). $n = 3$ mice at each time point. Staining of EdU was performed using a EdU cell proliferation assay kit based on the click chemistry technology according to the manufacture's instruction.

## Culture and functional assays of SSCs

Sorted SSCs were plated in culture plates coated with 0.1% (w/v) gelatin (G9391, Sigma, USA) for 1 h at room temperature before using. SSCs were cultured in α-MEM (Gibco, USA) with 10% fetal bovine serum (FBS) and 1% penicillin-streptomycin in hypoxia workstation (1% $O_2$ and 5% $CO_2$, Ruskinn Invivo2 400, UK). For colony-forming assay, 500 cells were plated in six-well plates and cultured for 14 days. Cell clones were stained with 0.1% crystal violet. For osteogenic differentiation, 1000 cells were plated in 24-well plate. After reaching confluency, osteogenic medium (containing 10 mmol/L β-glycerophosphate, 50 μg/mL ascorbic acid, and 10 nmol/L dexamethasone) were added for 21 days. The cells were fixed in 4% PFA for 10 min and stained with 0.2% alizarin red (A5533, Sigma) for 1 min. For chondrogenic differentiation, 20,000 cells were plated in high density or 1000 cells were plated in 24-well plate. The expanded cells were cultured in chondrogenic medium (containing sodium pyruvate, insulin, transferrin, sodium selenite, TGF-β, ascorbic acid, proline, and dexamethasone) for 21 days. The cells were fixed in 4% PFA for 10 min, acidized with 0.1 N HCl for 1 min, and stained with alcian blue (G1027, Servicebio, China) for 10 min. Percentages of positively stained areas were analyzed using the ImageJ software. For functional rescue assay, sorted SSCs were treated with 0.1 μmol SAG and simultaneously induced differentiation.

## Renal capsule transplantation

The renal capsule transplantation assay was performed as described (Chan et al, 2015). Briefly, sorted SSCs were sorted, resuspended in α-MEM (including 10% FBS) and pelleted, and then implanted underneath the renal capsule of 8–12-week-old immunodeficient NSG mice. Implanted cells developed into a graft after 4 weeks. The grafts were surgically removed for analysis.

## Plasmid construction and lentivirus transfection

Lentivirus pLKO.1 and retrovirus MSCV were used. For knock-down, the short hairpin RNAs (shRNA) of Mettl3 and Fem1b were cloned into pLKO.1-vector according to instructions. For over-expression experiment, mouse Gli1 and Mettl3 gene was amplified from cDNA obtained from SSCs and cloned into the MSCV vector. Lentivirus and retrovirus were produced by HEK293T cells transfected using calcium phosphate with virus packaging constructs pSPAX2 and pMD2.G or pCL-Eco. Viral supernatants were collected at 48 and 72 h after transfection, and filtered using a 0.45 μm low-protein binding membrane (Millipore, USA). The SSCs were infected with the specific virus in the presence of 8 μg/mL polybrene. Media were changed at 12 h after infection, and antibiotics (2 μg/mL puromycin) were added at 48 h after transfection if selection needed.

## RNA decay assay

Sorted SSCs were plated in 24-well plates and then treated with actinomycin D at a final concentration of 5 mg/mL for indicated time and collected. Total RNA was extracted by TRIzol and analyzed by RT-PCR. Gapdh was used as endogenous control. The half-life of mRNA was estimated according to previous study. The rate of disappearance of mRNA concentration at a given time ($dC/dt$) is proportional to both the rate constant for decay ($K_{decay}$) and the cytoplasmic concentration of the mRNA ($C$). This relation is described by the following equation: $dC/dt = -K_{decay}C$. The mRNA degradation rate $K_{decay}$ was estimated by: $\ln(C/C_0) = -K_{decay}t$. To determine the half-life ($t_{1/2}$), this means 50% of the mRNA is decayed ($C/C_0 = 1/2$). Substituted to the above equation got the following equation: $\ln(1/2) = -K_{decay} t_{1/2}$. from where: $t_{1/2} = \ln2/K_{decay}$.

## Quantitative qRT-PCR

Total RNA from SSCs and other target cells was extracted using TRIzol reagent (TAKARA) and RNA quality was assessed using NanoDrop 2000 (ThermoFisher Scientific). The ReverTraAce™ qPCR RT Kit (TOYOBO, Japan) was used to synthesize cDNA. The levels of target mRNAs were measured by SYBR Green mix using the CFX Opus 384 Real-Time PCR System (Bio-Rad, USA). Data were calculated using the $2^{-\triangle\triangle Ct}$ method and normalized with Gapdh and Actb. Primer sequences are listed in Appendix Table S1.

## MeRIP qRT–PCR

Methylated RNA immunoprecipitation (MeRIP)-qPCR was described as previous study (Yin et al, 2021). Quantification of mRNA from the MeRIP was carried out via $2^{-\triangle\triangle Ct}$ analysis against non-immunoprecipitated input RNA. All primer sequences can be found in Appendix Table S1.

## Western blot

The cell lysates or pull-downed proteins were separated by SDS-PAGE and transferred onto a polyvinylidene difluoride membrane. The membrane was blocked with 5% nonfat milk for 30 min at room temperature and incubated with the specified antibodies at 4 °C overnight. Membranes were subsequently incubated with HRP-conjugated secondary antibodies (Cell Signaling Technology) at room temperature for 1 h. The signals were detected by the luminescent image analyzer LAS-4000 mini (Fujifilm).

## Mass spectrometry

Flag-tagged Fem1b stably expressed in skeletal cells and HEK293T cells were harvested and lysed with IP buffer. After brief sonication, the lysates were centrifuged at 12,000 rpm for 10 min at 4 °C. The supernatants were immunoprecipitated with anti-Flag affinity gel and boiled in sample buffer. Eluted proteins were resolved by SDS-PAGE and visualized by Coomassie brilliant blue staining. Proteins in gels were then digested with trypsin in 50 mM ammonium bicarbonate at 37 °C overnight. After treatment with 5 mM DTT and 11 mM iodoacetamide, the resulting peptides were separated by silica capillary column and elute data at a flow rate of 0.3 μL/min with the UltiMate 3000 HPLC system (ThermoFisher Scientific) coupled with the Q Exactive mass spectrometer (ThermoFisher Scientific), which was set in the data-dependent acquisition mode using Xcalibur 2.2 software (ThermoFisher Scientific).

## m$^6$A dot plot

Total RNA was extracted from sorted cells. RNA samples were quantified by nanodrop and UV crosslinked to the Amersham Hybond-N + membrane (GE Healthcare, Cat#RPN203B), then RNA-bound membrane was blocked with 5% nonfat dry milk in PBS for 1 h at room temperature and incubated with a specific anti-m$^6$A antibody (Abcam, Cat#ab151230) overnight at 4 °C. HRP-conjugated secondary antibodies were added to the blots for 1 h at room temperature, and then the membrane was developed with enhanced chemiluminescence (Bio-Rad).

## SLIM-seq

For SLIM-seq (Yin et al, 2021), SSCs and other targets cells were sorted into TRIzol reagent (Takara) to extract RNA. m$^6$A antibody (ab15320, Abcam) and Protein A beads (ThermoFisher Scientific) were preincubated at 4 °C for 2 h. Then 100 ng total RNA per sample was incubated with the beads complex at 4 °C for another 2 h. Beads were washed twice with 200 μL IP buffer, twice with 200 μL low-salt buffer, twice with 200 μL high-salt buffer and once with 200 μL IP buffer. Captured RNA was eluted by heating beads for 2 min at 94 °C in 10 μL DEPC H$_2$O. The library was prepared using our customized protocol based on Smart-seq2. The input RNA and eluted RNA were reverse transcribed, pre-amplified and purified. Pre-amplified cDNA was fragmented by Tn5 enzyme to around 300 bp, followed by library generation using TruePrep® DNA Library Prep Kit V2 for Illumina Kit (Vazyme, TD501) and were pair-end sequenced on Illumina HiSeq X Ten sequencing platform.

## RNA-seq data analysis

Raw sequencing data processed by fastp (v. 0.20.1) for removing adapters and low-quality base pairs. Reads were aligned with HISAT2 (v. 2.2.1) to GRCm38 reference genome. Read counts were generated by featureCounts (v.2.0.1). Deseq2 (v1.36.0) was employed for data normalization and differential expression analysis of RNA-seq counts. Pathway analysis using GSEA (Gene Set Enrichment Analysis) software as used to identify functions of all differentially expressed genes. GO (Gene Ontology) and KEGG (Kyoto Encyclopedia of Genes and Genomes) analysis as used to identify functions of all differentially expressed genes (log$_2$ fold change >0.5 and $P < 0.05$).

## SLIMseq data analysis

All input and IP samples' raw sequencing data were preprocessed using fastp (v. 0.20.1), aligned with HISAT2 (v. 2.2.1) to GRCm38 reference genome, and generated read counts by featureCounts (v.2.0.1). For calculating the relative m$^6$A level for each gene, we used the DESeq2 to compare the read counts of genes between IP samples and input samples on a transcriptome-wide scale. Based on the adjusted fold change (Wald statistic, fourth column of output of DESeq2) of each gene from the output of DESeq2, we further calculated the Z-score of adjusted fold change in each cell type, and this Z-score represents the relative m$^6$A level. We identified high-confidence m$^6$A-tagged mRNAs by setting up the cutoff line based on log$_2$(fold change) >1 and $P$ value <0.05.

## Statistical analysis

Statistical analyses were performed using GraphPad Prism 10.0 software. Statistical significance was determined by unpaired two-sided Student's $t$ test, and one-way ANOVA. The sample sizes are stated in the figure legends to indicate biologically independent replicates used for statistical analyses. The $P$ values were labeled in the figures, and ns indicates no significance.

# Data availability

All RNA-seq data and SLIM-seq data reported in this paper is deposited in the public database GEO under the accession number GSE284224.

The source data of this paper are collected in the following database record: biostudies:S-SCDT-10_1038-S44318-025-00399-z.

# Peer review information

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

## Acknowledgements

We acknowledge the members of our laboratory for helpful discussion. We also thank all the staff in the core facility of Medical Research Institute at Wuhan University for their technical support. This work is supported by the grants from the National Key R&D Program of China (2022YFA0103200), and the National Natural Science Foundation of China (82325003, 82230007, 82200188, 82401137). This work is also supported by the Special Fund of China Postdoctoral Science Foundation (2022TQ0238) and the grants from the Fundamental Research Funds for the Central Universities (2042022dx0003, 2042024kf1022 and 2042021kf0225). We thank all the staff in the core facility of Medical Research Institute at Wuhan University for their technical support.

## Author contributions

**Jing Wang**: Conceptualization; Resources; Data curation; Software; Formal analysis; Investigation; Visualization; Methodology; Writing—original draft; Project administration. **Weidong Liu**: Resources; Software; Investigation; Methodology. **Tiantian Zhang**: Data curation; Software. **Manman Cui**: Methodology. **Kexin Gao**: Resources. **Pengbo Lu**: Resources; Investigation. **Shuxin Yao**: Resources; Methodology. **Ziyan Cao**: Resources; Methodology. **Yanbing Zheng**: Resources; Methodology. **Wen Tian**: Software; Investigation. **Yan Li**: Resources. **Rong Yin**: Resources; Software; Funding acquisition. **Jin Hu**: Data curation; Visualization; Methodology. **Guoqiang Han**: Data curation; Formal analysis; Validation. **Jianfei Liang**: Data curation; Formal analysis. **Fuling Zhou**: Resources. **Jihua Chai**: Supervision; Funding acquisition; Investigation; Project administration. **Haojian Zhang**: Conceptualization; Resources; Data curation; Formal analysis; Supervision; Funding acquisition; Investigation; Methodology; Writing—original draft; Project administration; Writing—review and editing.

Source data underlying figure panels in this paper may have individual authorship assigned. Where available, figure panel/source data authorship is listed in the following database record: biostudies:S-SCDT-10_1038-S44318-025-00399-z.

## Disclosure and competing interests statement

The authors declare no competing interests.

