## [Peer Review File · The EMBO Journal]

An epitranscriptomic program maintains skeletal stem cell quiescence via a METTL3-FEM1B-GLI1 axis

Haojian Zhang, Jing Wang, Weidong Liu, Tiantian Zhang, Manman Cui, Kexin Gao, Pengbo Lu, Shuxin Yao, Ziyao Cao, Yanbing Zheng, Wen Tian, Yan Li, Rong Yin, Jin Hu, Guoqiang Han, Jianfei Liang, Fuling Zhou, and Jihua Chai

Corresponding authors: Haojian Zhang (haojian_zhang@whu.edu.cn) , Jihua Chai (cjh@whu.edu.cn)

Review Timeline:

Submission Date:	28th Jul 24
Editorial Decision:	17th Sep 24
Revision Received:	23rd Dec 24
Editorial Decision:	29th Jan 25
Revision Received:	10th Feb 25
Accepted:	19th Feb 25

Editor: Ieva Gailite

Transaction Report:

Dear Dr. Zhang,

Thank you for submitting your manuscript for consideration by the EMBO Journal. I apologise for the protraction in the manuscript assessment process due to delays in referee report submission. We have now received comments from three reviewers, which are included below for your information.

As you can see, all reviewers are generally positive in their assessment, while raising a number of points that would need to be addressed and clarified before they can support publication. From my side, I find the raised points generally reasonable. In addition to the points by reviewers #1 and #2, adding more mechanistic insight into the role of FEM1B in direct regulation of GLL1 ubiquitination as requested by reviewer #3 would significantly strengthen the manuscript. Based on these positive reviewer assessments, I invite you to address their comments in a revised manuscript. I think that it would be useful to discuss the revision in more detail via email or phone/videoconferencing - please let me know which option you prefer.

We generally allow three months as standard revision time, which can be extended to six months in the case of major revisions. Should you foresee a problem in meeting this deadline, please let us know in advance to discuss an extension.

As a matter of policy, competing manuscripts published during this period will not negatively impact on our assessment of the conceptual advance presented by your study. However, please contact me as soon as possible upon publication of any related work to discuss the appropriate course of action.

When preparing your letter of response to the referees' comments, please bear in mind that this will form part of the Review Process File and will therefore be available online to the community. For more details on our Transparent Editorial Process, please visit our website: <https://www.embopress.org/page/journal/14602075/authorguide#transparentprocess>. Please also see the attached instructions for further guidelines on preparation of the revised manuscript.

Please feel free to contact me if have any further questions regarding the revision. Thank you for the opportunity to consider your work for publication, and I look forward to discussing your revision with you.

With best regards,

Ieva

- a point-by-point response to the referees' comments, with a detailed description of the changes made (as a word file).
- a word file of the manuscript text.
- individual production quality figure files (one file per figure)

- a complete author checklist, which you can download from our author guidelines (<https://www.embopress.org/page/journal/14602075/authorguide>).

- Expanded View files (replacing Supplementary Information)

We realize that it is difficult to revise to a specific deadline. In the interest of protecting the conceptual advance provided by the work, we recommend a revision within 3 months (16th Dec 2024). Please discuss the revision progress ahead of this time with the editor if you require more time to complete the revisions.

Referee #1:

The manuscript by Wang et al. reported an interesting phenomenon that at SSC or skeletal progenitor stages, m6A is mainly inherited from upstream development. However, significant changes in the m6A modification patterns in the downstream cells were found, implying that de novo m6A modifications have an essential role in the fate determination of skeletal cell hierarchy. The notion is further demonstrated using mouse genetic models and molecular biology experiments. Overall, the finding represents a new mechanism of the Mettl3-Femb2 axis in SSC fate determination. The experimental design and analysis are logical, and the results are presented clearly. However, a few points need to be addressed as below.

- 1) I could not find the profiling data in the GEO; please add the GEO numbers for better evaluation. It would be better if the DEG tables were attached as supplemental data.
- 2) The title can be more specific.
- 3) The most significant signaling changes were found in the Wnt and BMP, but not HH/Gli pathways from the profiling studies. It would be worthwhile to identify Mettl3 targets in Wnt and BMP pathways; or at least correlate the change of HH signaling with BMP and Wnt in this context.
- 4) How are the m6A status and total mRNA level of Gli1?
- 5) Fig.4H, please define how the 3'UTR of Femb1 mRNA is selected for luciferase assay. Does this plasmid-based mRNA overexpression achieve sufficient m6A modification to affect mRNA stability significantly? Does this process require other Mettl3 cofactors, which could be a rate-limiting factor?

Referee #2:

In Wang et al., the authors assess the role of Mettl3 and m6A modifications in skeletal stem cell biology. Overall, there is a high degree of novelty and interest in the findings. While regulation of quiescence has been shown to be important in other SSC types, there arguably has yet to be direct investigation of the role of quiescence in SSCs. Similarly, while there have been many studies of epigenetic regulation of bone formation, few to possibly none of these studies have directly investigated epigenetic regulation of SSCs. Thus, the present work is felt to have a high degree of novelty on several points. In general, the experimental methods for working with SSCs are seen as rigorous, though some areas for improvement are noted in the comments below. The biochemical mechanism is not investigated in an extremely high degree of depth, but this is seen as acceptable in the context of the overall manuscript, where cellular and phenotypic investigations are performed in more detail. The areas of concern identified below are largely seen as easily addressable, with major point #2 below deemed most important. Overall there is a high degree of enthusiasm for this manuscript if some of the below issues can be improved.

Major points

1. The CFU assay in Fig 3G-I is not really informative regarding SSC biology. CFU formation capacity is widely present in skeletal cells, including in non-SSCs and is essentially a measure of in vitro proliferation with unclear relationship to in vivo properties. This is also seen here, where the reduced CFU activity in SSCs contrasts with the increased fraction of cells entering

the cell cycle in vivo. It is recommended to remove this data or otherwise just clarify that the relationship of the CFU properties to in vivo biology of SSCs is unclear and that CFU formation is not an SSC specific property.

2. Upon reviewing Fig S1a, there is some concern that the FACS staining doesn't show the expected separation of populations. For instance, CD105 staining should indicate at least 2 populations, including clear negative and populations (albeit with some overlap between the two). The CD105 gating in S1a only shows a single population. Compare the CD105 signal here to similar plots in Chan et al. Cell 2015, Debnath et al. Nature 2015 and others. Similarly, how was the gate for CD200 determined? Further validation/justification of the gating strategy would increase confidence in the results.
3. Bone formation endpoints in renal capsule would be ideally measured quantitatively by uCT of the bone organoid to determine the total volume of bone formed. This is more reliable than histologic endpoints due to variability in histologic sampling (however the sections provided Fig 3k,l and 4m look fairly good in terms of comparability. If feasible, repeating one key renal capsule bone formation assay with a uCT endpoint would be desirable, though there is some flexibility on this. At the very least, lower power/uncropped images for the renal capsule histology should be provided.
4. Generally, stem cell quiescence is typically assayed by looking at label retention after a chase period, either after high level labeling with EdU with washout during the chase or with H2B-GFP label retention where H2B-GFP expression is suppressed during a chase phase. While there have arguably been no studies looking at quiescence in SSCs and therefore some flexibility in how this is conducted in the absence of a well-established standard, these kinds of label retention methods are generally preferred over cell cycle analysis for evaluating quiescence.
5. Bone cortical thickness should be provided in the uCT analysis in Fig 2. Some assessment of osteoclast numbers/activity is needed for completely skeletal phenotyping. At a minimum, TRAP staining or similar osteoclast staining should be performed.

Minor points 1. The abstract would be more approachable if the role of Mettl3 in m6A modification is mentioned upon first introduction of Mettl3.

2. In introducing Gli1, caution is required around claims that it marks osteogenic progenitors. This is true as stated, but an important point of confusion in the literature is that Gli1 is relatively widely expressed and cannot in and of itself define SSCs or other skeletal cell types. This is pertinent to the interpretation present story, as it cannot be excluded from the present study that the function of Mettl3-Fem1b-Gli1 signaling that is most important for bone homeostasis is actually downstream of the SSC itself.

3. Caution is needed regarding the diagram in Fig 1Aii. It is appreciated that this is largely based on the original Chan murine SSC report in Cell, however that study largely focused on establishing SSC identity and the evidence for the differentiation relationships linking the various non-stem populations is actually currently quite thin. It is recommended that the cells of interest just be listed in more of a tabular format without endorsing a specific differentiation hierarchy.

4. Cells with the Chan SSC immunophenotype can possibly include large numbers of relatively mature, likely contaminating osteolineage cells unless additional markers are added (Sun et al. Nature 2023). This should be considered and discussed.

Referee #3:

In this manuscript titled "Epitranscriptomic program maintains skeletal stem cell quiescence by regulating proteostasis", the authors demonstrated m6A program controls skeletal stem cell fate determination. First, they performed SLIM-seq to profile the m6A targets in 8 purified skeletal populations including stem and progenitor cells. The authors identified unique m6A programs during skeletal cell differentiation. Next, to study the function of m6A in skeletal development, the authors analyzed the Mettl3 f/f Col2a1 cre mice and demonstrated that METTL3 is critical for bone development. They found that METTL3 deficient skeletal stem cells are less quiescent and have reduced differentiation capacity. Mechanistically, the authors performed RNAseq in WT and METTL3 deficient skeletal stem cells and identified the E3 ligase FEM1B as a m6A target that is upregulated in METTL3 KO SSCs. They further showed that m6A controls RNA stability of Fem1b transcript and depletion of FEM1B reverses the SSC defect upon loss of m6A. Gli1 is a transcription factor that is critical for SSCs and its expression is controlled by FEM1B. Furthermore, Gli1 protein ubiquitination and protein degradation is enhanced upon m6A loss when there is increased FEM1B. To further test whether Gli1 is a substrate of FEM1B. Lastly, the authors showed that by activating hedgehog signaling using SAG reversed the SSC defect upon loss of m6A.

Overall, the authors identified the m6A-FEM1B-Gli1 axis that is critical in controlling SSC function. They provide an extensive set of data and will be an important resource on m6A profiles in skeletal hierarchy. Moreover, their mechanistic studies identifying the m6A-FEM1B-Gli1 pathway is also an important finding. However, key experiments are needed to support the claim of the work.

1. There seems to be a jump from the identification of FEM1B as a m6A target in SSC to Gli1. Given FEM1B is a E3 ligase, the authors need to provide the following data:
 - a. Is the ubiquitination of Gli1 dependent on FEM1B?

b. Does FEM1B depletion reverse the increased ubiquitination and degradation of GLI1?

2. It appears that the "proteasome mediated protein degradation" pathway is one of the key m6A controlled process in SSCs. And the authors identify the E3 ligase FEM1B as the key target in the later part of the manuscript. The authors only show an increased protein of FEM1B and reduced GLI1. It is not clear exactly how they focused in on FEM1B. Additionally, FEM1B has many substrates how is that this directly links to GLI1. Can the authors provide a less biased way to determine FEM1B substrates or provide a more balanced way to explain their linear rational.

3. In Fig 5 the authors showed that agonist of Hedgehog signaling SAG reverse m6A deficient SSC phenotype. Are any players of hedgehog signaling, including Gli1, direct m6A targets in SSC? How does SAG impact FEM1B level? Does SAG treatment increase Gli1 transcription? Does GLI1 overexpression phenocopy SAG treatment?

Point-by-point response letter

Comments:

Referee #1:

The manuscript by Wang et al. reported an interesting phenomenon that at SSC or skeletal progenitor stages, m6A is mainly inherited from upstream development. However, significant changes in the m6A modification patterns in the downstream cells were found, implying that de novo m6A modifications have an essential role in the fate determination of skeletal cell hierarchy. The notion is further demonstrated using mouse genetic models and molecular biology experiments. Overall, the finding represents a new mechanism of the Mettl3-Fem1b axis in SSC fate determination. The experimental design and analysis are logical, and the results are presented clearly. However, a few points need to be addressed as below.

Response: We do appreciate this very positive and inspiring comment of the reviewer. And we also thank you for your all the great suggestions. As described in the following (also shown in this revised manuscript), we have addressed all the concerns and provided all the new data for the detailed mechanisms. We hope you are satisfied with our efforts in revising this manuscript.

1) I could not find the profiling data in the GEO; please add the GEO numbers for better evaluation. It would be better if the DEG tables were attached as supplemental data.

Response: We appreciate this suggestion, and upload the sequencing data to GEO. The GEO number is GSE284224. Meanwhile, the related DEG tables are also attached in the appendix datasets in this revised manuscript.

2). The title can be more specific.

Response: We appreciate this point the reviewer raised. In this revised manuscript, we follow this suggestion and change the title to be more specific. The new title is “Epitranscriptomic program maintains skeletal stem cell quiescence via METTL3-FEM1B-GLI1 axis”.

3) The most significant signaling changes were found in the Wnt and BMP, but not HH/Gli pathways from the profiling studies. It would be worthwhile to identify Mettl3 targets in Wnt and BMP pathways; or at least correlate the change of HH signaling with BMP and Wnt in this context.

Response: We do appreciate this point you raised.

1) We agree with you, and find that multiple targets involved in Wnt signaling pathway shows significant change from the profiling data. As the role of Wnt signaling pathway in bone development has been well investigated previously (*Chan, Charles K F et al. Cell. 2015; Wasnik, Samiksha et al. Science Advances. 2019; Matsushita, Yuki et al. Nature Communications. 2020*), we did not focus on this pathway in this study.

2) By integrating the DEGs of RNA-seq data and SLIM-seq data, we observed BMP signaling pathway related genes (e.g. *Bmpr1a* and *Bglap*) and found that their mRNA and m6A levels

were decreased after *Mettl3* deletion in SSCs, suggesting that BMP signaling pathway is regulated by RNA m6A modification in SSCs. We added this new information in this revised manuscript (**new Appendix Fig. S5A,B**).

4) How are the m6A status and total mRNA level of *Gli1*?

Response: We do appreciate this point you raised. Indeed, we analyzed *Gli1* mRNA level in sorted skeletal stem cells (SSCs) from WT and *Mettl3*^{KO} mice, and did not observe obvious difference between WT and *Mettl3*^{KO} (**new Appendix Fig. S6E**), suggesting that *Gli1* is not directly regulated at the mRNA level via m6A modification. Through analyzing MeRIP-sequencing data (Wu, Yunshu *et al. Nature Communications. 2018*), we also did not observe obvious m6A peak on *Gli1* mRNA, as showed by IGV (**new Appendix Fig. S6F**). Thus, these data suggest that *Gli1* is not a direct m6A target.

5) Fig.4H, please define how the 3'UTR of *Fem1b* mRNA is selected for luciferase assay.

Response: We appreciate this point you raised. To define the 3'UTR region of *Fem1b* used in the luciferase reporter for examining the effect of m6A modification, **1)** we first analyzed MeRIP-seq data and observed a significant m6A peak in the 3'UTR of *Fem1b* (**Fig. 4E**), which indicates an enrichment of m6A modification in this 3'UTR region of *Fem1b*. **2)** Next, focusing on this 3'UTR region, we performed m6A RIP-PCR and found that m6A modification was significant enriched in this region, and this enrichment of m6A modification was abrogated upon *Mettl3* deletion (**Fig. 4F**). Thus, these data confirmed the m6A modification in this 3'UTR region of *Fem1b*. **3)** Therefore, we chose this m6A-enriched

Fem1b 3'UTR region (~450bp) to construct the firefly luciferase reporter. We integrated a schematic model of this luciferase reporter and described it in this revised manuscript (Fig. 4H).

Does this plasmid-based mRNA overexpression achieve sufficient m⁶A modification to affect mRNA stability significantly? Does this process require other Mettl3 cofactors, which could be a rate-limiting factor?

Response: Thanks for this great point you raised. 1) Following your suggestion, we first assessed whether Mettl3 overexpression increases the m⁶A level of Fem1b 3'UTR used in this reporter plasmid. As expected, m⁶A RIP-PCR assay showed significantly increase of m⁶A modification accompanied with a decrease in the luciferase activity upon Mettl3 overexpression, which indicates that the plasmid-based overexpression could effectively achieve m⁶A modification to affect mRNA stability (Fig. 4H; Appendix Fig. S5C). These data validated the accuracy of this reporter

2) It is known that Mettl3 is the core catalytic unit of RNA m⁶A methyltransferase complex that is a multicomponent complex composed of many cofactors, such as Mettl14, Wtap, Virma, RBM15/15B and ZC3H13. To address whether the Mettl3 cofactor are required in the system we used in this study, we chose METTL14, another core subunit of methyltransferase complex. As expected, we found that knockdown of Mettl14 blocked the decrease of luciferase activity induced by Mettl3 overexpression, indicating that Mettl14 is required in this system (Panel A, shown below).

Referee #2:

In Wang et al., the authors assess the role of Mettl3 and m6A modifications in skeletal stem cell biology. Overall, there is a high degree of novelty and interest in the findings. While regulation of quiescence has been shown to be important in other SSC types, there arguably has yet to be direct investigation of the role of quiescence in SSCs. Similarly, while there have been many studies of epigenetic regulation of bone formation, few to possibly none of these studies have directly investigated epigenetic regulation of SSCs. Thus, the present work is felt to have a high degree of novelty on several points. In general, the experimental methods for working with SSCs are seen as rigorous, though some areas for improvement are noted in the comments below. The biochemical mechanism is not investigated in an extremely high degree of depth, but this is seen as acceptable in the context of the overall manuscript, where cellular and phenotypic investigations are performed in more detail. The areas of concern identified below are largely seen as easily addressable, with major point #2 below deemed most important. Overall there is a high degree of enthusiasm for this manuscript if some of the below issues can be improved.

Response: We thank the reviewer's expert comments. We also do appreciate all the great suggestions the reviewer raised, and following these suggestions, we have addressed all the concerns and provided new data for the detailed mechanisms. We hope the reviewer could be satisfied with our efforts.

Major points

1. The CFU assay in Fig 3G-I is not really informative regarding SSC biology. CFU formation capacity is widely present in skeletal cells, including in non-SSCs and is essentially a measure of in vitro proliferation with unclear relationship to in vivo properties. This is also seen here, where the reduced CFU activity in SSCs contrasts with the increased fraction of cells entering the cell cycle in vivo. It is recommended to remove this data or otherwise just clarify that the relationship of the CFU properties to in vivo biology of SSCs is unclear and that CFU formation is not an SSC specific property.

Response: Thanks for this great point you raised, and we totally agree with you that CFU is not an ideal assay to test SSC properties. Following your suggestion, we remove the CFU data in this revised manuscript.

2. Upon reviewing Fig S1a, there is some concern that the FACS staining doesn't show the expected separation of populations. For instance, CD105 staining should indicate at least 2 populations, including clear negative and populations (albeit with some overlap between the two). The CD105 gating in S1a only shows a single population. Compare the CD105 signal here to similar plots in Chan et al. Cell 2015, Debnath et al. Nature 2015 and others. Similarly, how was the gate for CD200 determined? Further validation/justification of the gating strategy would increase confidence in the results.

Response: We do appreciate this point you raised. **1)** We think you bring an important issue about how the gates are set up when performing multicolor flow cytometry. In our study, we usually use fluorescence minus one (FMO) control to determine where the gates should be set. As shown below (**Panel A and B**), we built the FMO controls for CD105-FITC and CD200-APC that help us to clearly determine and gate the negative and positive CD105 or

CD200 populations. **2)** We also followed your suggestion and compared the plots from the two reference papers you mentioned. As shown below (**Panel C**), we used the contour plot (the same strategies used in the studies you referred to) to display our flow cytometry data, and we gated the negative and positive CD105 or CD200 populations based on the FMO controls. We also replaced **Figure S1A** with the new contour plot in this revised manuscript.

3. Bone formation endpoints in renal capsule would be ideally measured quantitatively by μ CT of the bone organoid to determine the total volume of bone formed. This is more reliable than histologic endpoints due to variability in histologic sampling (however the sections provided Fig 3k,l and 4m look fairly good in terms of comparability). If feasible, repeating one key renal capsule bone formation assay with a μ CT endpoint would be desirable, though there is some flexibility on this. At the very least, lower power/uncropped images for the renal capsule histology should be provided.

Response: We thank for this great point you raised. Following your suggestions, **1)** we performed the renal capsule bone formation assay and measured quantitatively by μ CT to determine the total volume of bone formed *in vivo*. As expected, through analyzing by μ CT, we confirmed that *Mettl3* deletion impaired the bone formation capacity of SSCs (**new Figure 3K,L**). **2)** We also provided the uncropped images for the renal capsule histology (**new Appendix Fig. S4A,B**).

3) In term of Figure 4M, we also performed μ CT to measure the bone formation beneath the renal capsule after knocking down *Fem1b* expression in *Mettl3*^{KO} SSCs. As expected, μ CT showed that *Fem1b* knockdown could markedly rescue *in vivo* bone formation ability of *Mettl3*^{KO} SSCs (**new Fig. 4N,O in this revised manuscript**). Meanwhile, we also showed the uncropped images of the renal capsule histology in this revised manuscript (**new Appendix Fig. S5D**).

4. Generally, stem cell quiescence is typically assayed by looking at label retention after a chase period, either after high level labeling with EdU with washout during the chase or with H2B-GFP label retention where H2B-GFP expression is suppressed during a chase phase. While there have arguably been no studies looking at quiescence in SSCs and therefore some flexibility in how this is conducted in the absence of a well-established standard, these kinds of label retention methods are generally preferred over cell cycle analysis for evaluating quiescence.

Response: We do appreciate this great suggestion you raised. We totally agree with you that labeling-retention cell (LRC) assay using EdU labeling or H2B-GFP labeling is better for evaluating stem cell quiescence. As you mentioned, there have been no studies on the quiescence of SSCs. In this revised manuscript, following your suggestion, we tried our best to examine the cell cycle of SSCs by EdU labeling. EdU was administered into mice twice with 3-hour interval, mice were euthanized at six hours. We observed more EdU labeling in the epiphyseal and growth plate area of femurs of *Mettl3^{KO}* mice (**new Fig. 3G in this revised manuscript**), indicating that more *Mettl3^{KO}* SSCs are in the cycling phase. This data is consistent with the cell cycle analysis (**Fig. 3E,F**).

5. Bone cortical thickness should be provided in the uCT analysis in Fig 2.

Response: We appreciate this suggestion you provided. In this revised manuscript, we analyzed the bone cortical thickness using μ CT. Interestingly, there were no significant differences in the bone cortical thickness (**new Appendix Fig. S2P,Q in this revised manuscript**). This might be due to Col2a1-cre mice model, given that Col2a1 mainly marks skeletal stem and progenitors in the growth plate but not osteoblast in the cortical area. Thus, it is reasonable to see no difference in the cortical thickness between WT and *Mettl3^{fl/fl};Col2a1-Cre* mice.

Some assessment of osteoclast numbers/activity is needed for completely skeletal phenotyping. At a minimum, TRAP staining or similar osteoclast staining should be performed.

Response: We appreciate this great point you raised. Following this suggestion, we performed TRAP staining to assess osteoclast, and found there was no significant difference in osteoclast activity (**new Appendix Fig. S2F in this revised manuscript**).

Minor points

1. The abstract would be more approachable if the role of *Mettl3* in m6A modification is mentioned upon first introduction of *Mettl3*.

Response: We appreciate this suggestion, and described *Mettl3* function in the abstract in this revised manuscript.

2. In introducing Gli1, caution is required around claims that it marks osteogenic progenitors. This is true as stated, but an important point of confusion in the literature is that Gli1 is relatively widely expressed and cannot in and of itself define SSCs or other skeletal cell types. This is pertinent to the interpretation present story, as it cannot be excluded from the present study that the function of Mettl3-Feb1b-Gli1 signaling that is most important for bone homeostasis is actually downstream of the SSC itself.

Response: We appreciate this great suggestion. 1) We remove the description about “Gli1 marks osteogenic progenitors”. 2) We discussed this point in the Discussion section (Page 16) of this revised manuscript.

3. Caution is needed regarding the diagram in Fig 1Aii. It is appreciated that this is largely based on the original Chan murine SSC report in Cell, however that study largely focused on establishing SSC identity and the evidence for the differentiation relationships linking the various non-stem populations is actually currently quite thin. It is recommended that the cells of interest just be listed in more of a tabular format without endorsing a specific differentiation hierarchy.

Response: We appreciate this suggestion, and totally agree with you. Following your suggestion, we list these cell types in a tabular format in this revised manuscript (**new Fig. 1B**).

Figure 1B

Cell type	Markers	Numbers of m ⁶ A-tagged genes
SSC	CD45 ⁻ Ter119 ⁻ Tie2 ⁻ AlphaV ⁺ Thy1 ⁺ 6C3 ⁻ CD105 ⁺ CD200 ⁺	1909
p-BCSP	CD45 ⁻ Ter119 ⁻ Tie2 ⁻ AlphaV ⁺ Thy1 ⁺ 6C3 ⁻ CD105 ⁺ CD200 ⁻	2101
BCSP	CD45 ⁻ Ter119 ⁻ Tie2 ⁻ AlphaV ⁺ Thy1 ⁺ 6C3 ⁻ CD105 ⁺	1333
PCP	CD45 ⁻ Ter119 ⁻ Tie2 ⁻ AlphaV ⁺ Thy1 ⁺ 6C3 ⁻ CD105 ⁺ CD200 ⁻	2170
THY	CD45 ⁻ Ter119 ⁻ Tie2 ⁻ AlphaV ⁺ Thy1 ⁺ 6C3 ⁻ CD105 ⁺ CD200 ⁻	2172
BLSP	CD45 ⁻ Ter119 ⁻ Tie2 ⁻ AlphaV ⁺ Thy1 ⁺ 6C3 ⁻ CD105 ⁺	2252
6C3	CD45 ⁻ Ter119 ⁻ Tie2 ⁻ AlphaV ⁺ Thy1 ⁺ 6C3 ⁺ CD105 ⁻	2533
HEC	CD45 ⁻ Ter119 ⁻ Tie2 ⁻ AlphaV ⁺ Thy1 ⁺ 6C3 ⁺ CD105 ⁻	2179

4. Cells with the Chan SSC immunophenotype can possibly include large numbers of relatively mature, likely contaminating osteolineage cells unless additional markers are added (Sun et al. Nature 2023). This should be considered and discussed.

Response: We appreciate this suggestion. We discussed this point in the Discussion section (Page 15) of this revised manuscript, as shown below:

“It should be noted that SSCs is defined using the immunophenotype markers as previous study (Chan, Charles K F et al. Cell. 2015). Although we cannot exclude the contamination of osteolineage cells in SSC population, however, we do not believe this issue could affect the whole m⁶A landscape of the skeletal hierarchy we observed.”

Referee #3:

In this manuscript titled "Epitranscriptomic program maintains skeletal stem cell quiescence by regulating proteostasis", the authors demonstrated m6A program controls skeletal stem cell fate determination. First, they performed SLIM-seq to profile the m6A targets in 8 purified skeletal populations including stem and progenitor cells. The authors identified unique m6A programs during skeletal cell differentiation. Next, to study the function of m6a in skeletal development, the authors analyzed the *Mettl3* f/f *Col2a1* cre mice and demonstrated that METTL3 is critical for bone development. They found that METTL3 deficient skeletal stem cells are less quiescent and have reduced differentiation capacity. Mechanistically, the authors performed RNAseq in WT and METTL3 deficient skeletal stem cells and identified the E3 ligase FEM1B as a m6A target that is upregulated in METTL3 KO SSCs. They further showed that m6A controls RNA stability of *Fem1b* transcript and depletion of FEM1B reverses the SSC defect upon loss of m6A. *Gli1* is a transcription factor that is critical for SSCs and its expression is controlled by FEM1B. Furthermore, *Gli1* protein ubiquitination and protein degradation is enhanced upon m6A loss when there is increased FEM1B. To further test whether *Gli1* is a substrate of FEM1B. Lastly, the authors showed that by activating hedgehog signaling using SAG reversed the SSC defect upon loss of m6A. Overall, the authors identified the m6A-FEM1B-*Gli1* axis that is critical in controlling SSC function. They provide an extensive set of data and will be an important resource on m6A profiles in skeletal hierarchy. Moreover, their mechanistic studies identifying the m6A-FEM1B-GLI1 pathway is also an important finding. However, key experiments are needed to support the claim of the work.

Response: We do appreciate this very positive and inspiring comment. We also thank all the great suggestions you raised, which definitely improve this work. As described in the following and also shown in this revised manuscript, we have provided new data for the detailed mechanisms and addressed all the concerns. We hope you could be satisfied with our efforts.

1. There seems to be a jump from the identification of FEM1B as a m6A target in SSC to GLI1. Given FEM1B is a E3 ligase, the authors need to provide the following data:

- a. Is the ubiquitination of GLI1 dependent on FEM1B?
- b. Does FEM1B depletion reverse the increased ubiquitination and degradation of GLI1?

Response: Thanks for this great point you raised.

1) We have observed that an increase of the *Gli1* ubiquitination is associated with higher expression of *Fem1b* in *Mettl3*^{KO} SSCs (Fig. 5A,B).

2) Following your suggestion, to further examine whether the ubiquitination of *Gli1* is dependent on *Fem1b*, we knocked down *Fem1b* in SSCs. As expected, the ubiquitination of *Gli1* was decreased upon *Fem1b* deletion, which accompanied with increased expression of *Gli1* (new Appendix Fig. S6G,H in this revised manuscript).

3) We also deleted *Fem1b* in *Mettl3*^{KO} cells to assess whether *Fem1b* depletion reverses the increased ubiquitination and degradation of *Gli1*. As expected, the ubiquitination level and degradation of *Gli1* was obviously reversed (new Appendix Fig. S6I,J in this revised manuscript).

Together, these data indicate that Fem1b regulates Gli1 protein stability by modifying its ubiquitination.

2. It appears that the "proteasome mediated protein degradation" pathway is one of the key m⁶A controlled process in SSCs. And the authors identify the E3 ligase FEM1B as the key target in the later part of the manuscript. The authors only show an increased protein of FEM1B and reduced GLI1. It is not clear exactly how they focused in on FEM1B.

Additionally, FEM1B has many substrates how is that this directly links to GLI1. Can the authors provide a less biased way to determine FEM1B substrates or provide a more balanced way to explain their linear rational.

Response: We thank this point you raised. We also apologized for unclear description about how we chose Fem1b. Indeed, Fem1b was selected basing on the following reasons: 1) By analyzing the m⁶A profiling across skeletal hierarchy, we found that genes enriching in proteasome-mediated ubiquitin-dependent protein catabolic process showed higher m⁶A modifications and lower mRNA levels, indicating that this pathway is one key m⁶A-controlled process in SSCs. 2) To identify the potential targets, we performed RNA-seq and SLIM-seq to globally and unbiasedly analyze gene expression in *Mettl3*^{KO} SSCs. We integrated these transcriptomics data with m⁶A profiles data, and we identified Fem1b and Ube2h involving in protein ubiquitination. Given that Fem1b is the substrate-recognition component of a Cul2-RING (CRL2) E3 ubiquitin-protein ligase complex, we decided to choose Fem1b for further investigation. We restated this point in this revised manuscript.

3) To identify the potential substrates of Fem1b, we expressed Flag-tagged Fem1b

(Flag-Fem1b) in 293T cells and skeletal cells and performed liquid chromatograph-mass spectrometry (LC-MS) analysis. A total of 127 and 96 proteins in skeletal cells and 293T cells respectively were identified by LC-MS analysis. Integrating these two datasets, we identified 16 overlapped proteins (e.g. Col1a1, Gli1, Myh9, Pdik11) that were pulled down in both cell types, which served as potential substrates of Fem1b in both skeletal cells and 293T (**new Appendix Fig. S6A,B in this revised manuscript**). Interestingly, among these potential substrates, Gli1 was one of the top candidates (**new Appendix Fig. S6C in this revised manuscript**). Thus, we focused on Gli1 and conducted co-IP assay. As expected, we confirmed the interaction between Fem1b and Gli1 (**new Appendix Fig. S6D in this revised manuscript**). Therefore, these data prompted us to focus on Gli1 as a downstream substrate of Fem1b.

3. In Fig 5 the authors showed that agonist of Hedgehog signaling SAG reverse m6A deficient SSC phenotype. Are any players of hedgehog signaling, including Gli1, direct m6A targets in SSC?

Response: Thanks for this great point you raised. Indeed, we analyzed Gli1 mRNA level in sorted skeletal stem cells (SSCs) from WT and *Mettl3*^{KO} mice, and did not observe obvious difference between WT and *Mettl3*^{KO} (**new Appendix Fig. S6E in this revised manuscript**), suggesting that Gli1 is not directly regulated at the mRNA level via m6A modification. Through analyzing MeRIP-sequencing data (Wu, Yunshu et al. Nature communications. 2018), we also did not observe obvious m6A peak on Gli1 mRNA, as showed by IGV (**new Appendix Fig. S6F in this revised manuscript**). Thus, these data suggest that Gli1 is not a direct m6A target.

We also checked other players in Hedgehog pathway, such as *Ihh*, *Dhh*, *Shh*, *Gli2* and *Gli3*. Similarly, there were no significant m6A peaks in IGV, and their mRNA levels were also not significantly changed (**Panel A-B, shown below**). Together, these results suggest that Hedgehog genes are not directly regulated by m6A.

Figure for reviewers removed.

How does SAG impact FEM1B level? Does SAG treatment increase *Gli1* transcription?

Response: Thanks for this point you raised. Following your suggestion, we used SAG to treat SSCs, and found that SAG treatment increased *Gli1* mRNA level in a dose-dependent manner. Interestingly, we did not observe significant change in *Fem1b* mRNA level (**new Appendix Fig. S6M,N in this revised manuscript**). These data suggest that SAG treatment can increase *Gli1* transcription but not FEM1B expression.

Does GLI1 overexpression phenocopy SAG treatment?

Response: Thanks for this great point you raised. Following your suggestion, we overexpressed *Gli1* in *Mettl3^{KO}* SSCs and performed osteogenesis *in vitro*. As expected, we found that *Gli1* overexpression rescued the defective osteogenic differentiation ability of *Mettl3^{KO}* SSCs, which is in line with the phenotypes with SAG treatment (**new Appendix Fig. S6K,L in this revised manuscript**).

Dear Dr. Zhang,

Thank you for submitting a revised version of your manuscript. We have now received input from all original reviewers, who now find that their main concerns have been addressed satisfactorily and recommend acceptance of the manuscript. Therefore I would be happy to accept the manuscript for publication after a textual revision in response to the remaining textual point by reviewer #2.

Additionally, there remain a few editorial points that need addressing before I can extend official acceptance of the manuscript:

1. Please check that the funding information is correct and identical both in the manuscript and our online system. All funders need to be included in our online form as separate funding sources. Currently, only a single funder and grant number are submitted in our online system.
2. Please make sure that the order of the sections in the manuscript is as follows: Abstract / Keywords / Introduction / Results / Discussion / Methods / Acknowledgments / Disclosure and competing interests statement / References / Figure legends / Tables and their legends / Expanded View Figure legends.
3. Please reduce the number of keywords to five.
4. Please rename "Conflict of interest" section into "Disclosure and competing interests statement" (further info: <https://www.embopress.org/page/journal/14602075/authorguide#conflictsofinterest>).
5. CRedit has replaced the traditional author contributions section because it offers a systematic, machine-readable author contributions format that allows for more effective research assessment. Please remove the Authors Contributions from the manuscript and use the free text boxes beneath each contributing author's name in our online submission system to add specific details on the author's contribution. More information is available in our guide to authors.
6. In the Data Availability section, please add a resolvable link to the GSE284224 dataset. More information about the format of this section can be found here: <https://www.embopress.org/page/journal/14602075/authorguide#dataavailability>.
7. In the Appendix, the figure and table nomenclature should be consistently used as follows: Appendix Figure S1 - S6 and Appendix Table S1. add the manuscript title in the front page and include page numbers in the table of contents.
8. Please update references according to The EMBO Journal style - where there are more than 10 authors on a paper, the first 10 should be listed, followed by 'et al.' Please see further information here: <https://www.embopress.org/page/journal/14602075/authorguide#referencesformat>
9. In our standard image integrity check, we noted that the following figure panels are reused in the manuscript:
 - between Figure 3 I & J and Appendix Figure S4 A & B;
 - between Figure 4 M and Appendix Figure S5 D.If this was intentional, please mention the reuse in the figure legend.
10. During our routine text plagiarism check, we noted that numerous sentences in the introduction section of the manuscript show high similarity to those from other publications, including those from your group. Please rephrase the text accordingly. I have attached screenshots with a couple of instances that in particular would need rephrasing.
11. Our data editors have flagged the following issues in figure legends that need correcting:
 - Please provide the exact p values in the legends of figures 1G-J; 2D, F, G, I, K, L, M, N; 3C, D, F, L; 4C, F, H, I, J, L, O; 5F, I, K; supplementary figure(s) 1B, H, G; 2A, J-O; 3C, 4F, G, I, K; 5A, B, C, G; 6M.
 - Please indicate the statistical test used for data analysis in the legends of figures 1E, 4A, B
 - Please note that in figure 3D there is a mismatch between the annotated p values in the figure legend and the annotated p values in the figure file that should be corrected.
 - Please define the box plots in terms of minima, maxima, centre, bounds of box and whiskers, and percentile in the legends of figure 1C and supplementary figure 1E.
 - Please provide information on the number and nature of replicates in the legends of figures 1C, G-J; 4A, C, F, G, H, I, J, L; 5F; supplementary figure(s) 1B, E; 2A, 3C, 4F, G, I, K; 5A, B, C, G.
 - Please define the scale bar for supplementary figures 2P, Q
12. Papers published in The EMBO Journal are accompanied online by a 'Synopsis' to enhance discoverability of the manuscript. It consists of A) a short (1-2 sentences) summary of the findings and their significance, B) 3-4 bullet points highlighting key results and C) a synopsis image that is 550x300-600 pixels large (width x height, jpeg or png format). You can either show a model or key data in the synopsis image. Please note that the image size is rather small and that text needs to be readable at the final size. Please send us this information together with the revised manuscript.

With best wishes,

leva

We realize that it is difficult to revise to a specific deadline. In the interest of protecting the conceptual advance provided by the work, we recommend a revision within 3 months (29th Apr 2025). Please discuss the revision progress ahead of this time with the editor if you require more time to complete the revisions.

Referee #1:

All comments are addressed with extensive supplemental data in good quality, and I recommend acceptance to publication.

Referee #2:

The authors have responded thoroughly to the comments raised on the initial submission and essentially all points have been strongly addressed. Only one suggested textual change remains in response to original reviewer 2, comment 5 response from the authors:

Generally col2a1-cre is observed to delete widely in almost all skeletal tissue, including in periosteal/cortical osteoblasts, not only in growth plate populations as was claimed. Many papers have documented this, but one example using inducible cre systems that comes to mind is in Ono et al. Nat Cell Biol 2014. Therefore the absence of a cortical phenotype may indicate that the Mettl3 pathway is not active in periosteal or endocortical cells or that compensation is masking observing a phenotype. This should be acknowledged and commented upon in the text.

Referee #3:

The authors have responded to all the concerns with more experiments and improved explanations.

Point-by-point response letter

Referee #1:

All comments are addressed with extensive supplemental data in good quality, and I recommend acceptance to publication.

Response: We appreciate that the reviewer is satisfied with our efforts.

Referee #2:

The authors have responded thoroughly to the comments raised on the initial submission and essentially all points have been strongly addressed. Only one suggested textual change remains in response to original reviewer 2, comment 5 response from the authors:

Generally col2a1-cre is observed to delete widely in almost all skeletal tissue, including in periosteal/cortical osteoblasts, not only in growth plate populations as was claimed. Many papers have documented this, but one example using inducible cre systems that comes to mind is in Ono et al. Nat Cell Biol 2014. Therefore the absence of a cortical phenotype may indicate that the Mettl3 pathway is not active in periosteal or endocortical cells or that compensation is masking observing a phenotype. This should be acknowledged and commented upon in the text.

Response: We appreciate the reviewer is satisfied with our efforts, and also thank this last suggestion about the textual change. In the revised manuscript, we acknowledged and discussed it in the Discussion section in Page 17, as shown in the following:

“In addition, previous fate-mapping studies showed that cells expressing Col2-cre recombinase contribute to multiple cell types including chondrocytes, perichondrial precursors, osteoblasts, and stromal cells in the skeletal system. Thus, it is reasonable that Mettl3 deletion in these different cells may result in compensatory phenotypes. This might explain our observation that the bone cortical thickness did not change upon Mettl3 deficiency.”

Referee #3:

The authors have responded to all the concerns with more experiments and improved explanations.

Response: We appreciate the reviewer is satisfied with our efforts.

Dear Dr. Zhang,

Thank you for addressing the final editorial issues. I am now pleased to inform you that your manuscript has been accepted for publication.

Before we forward your manuscript to our publishers, I would like to propose some edits in the manuscript title, abstract and synopsis (please also see the attached file). I have also written a short blurb that will accompany the title of your manuscript in our online table of contents. Please take a look and let me know if any corrections are needed.

Title:

An epitranscriptomic program maintains skeletal stem cell quiescence via a METTL3-FEM1B-GLI1 axis

Blurb:

The landscape of RNA m6A methylation across murine skeletal cell populations reveals the role of METTL3 in skeletal stem cell maintenance.

Synopsis:

m6A RNA modification has been linked to regulation of osteogenic differentiation. This study defines the RNA m6A methylome of skeletal cell populations and reveals the role of METTL3-dependent regulation of FEM1B-GLI1 module in maintaining the function of skeletal stem cells (SSCs).

- Super-low-input m6A sequencing (SLIM-seq) reveals a comprehensive m6A landscape in murine skeletal cell populations.
- Mettl3 deletion impairs bone development.
- METTL3 is required for maintaining skeletal stem cell quiescence and multipotency.
- m6A modification promotes FEM1B mRNA degradation and subsequently increases GLI1 protein stability in SSCs.

Finally, we would like to promote your manuscript among the Chinese readership. Therefore, we would like to invite you to prepare a short summary of the manuscript in Chinese (1500-2000 Chinese characters), which we will promote on the WeChat platform 'BioArt' with more than 610,000 followers.

If you are interested in this opportunity, we recommend covering the article very close to its online publication date. Thus, ideally we would very much appreciate if you could send us a draft within the next 7 working days. Please let us know whether or not you would be interested in contributing such a short summary in Chinese.

I have included below some general guidelines on how to prepare a summary and a link to recent examples for your reference.

If you have any questions, please do not hesitate to contact the Editorial Office. Thank you for this contribution to The EMBO Journal and congratulations on a nice study!

With best wishes,

Ieva

Ieva Gailite, PhD
Senior Scientific Editor
The EMBO Journal
Meyerhofstrasse 1
D-69117 Heidelberg
Tel: +4962218891309

General WeChat Summary Guidelines

1. These summary articles are meant to be targeting general audience so please limit the use of specialized technical terms, acronyms and jargon.
2. A summary usually starts with brief background information of the reported work, which is followed by explaining the findings in some detail, and ends with a short review of the conclusions as well as the implications of the work and future directions for the research.
3. The summary should at least contain one graphical item, such as a scheme or a figure from the paper.
4. Please provide ONE SINGLE document containing all text and graphical materials, ideally as a Word.docx or .doc file. Please DO NOT provide the document as a .pdf file.
5. Please DO NOT publicly release the document before the paper is officially published online.

Summary Examples

EMBO J | 罗招庆/欧阳松应揭示谷酰胺脱氨酶MvcA的去泛素化功能

EMBO J | 王松灵院士团队揭示组织内应力调控大型哺乳动物乳恒牙替换的新机制
